# Feature analysis of joint motion in paralyzed and non-paralyzed upper limbs while reaching the occiput: A cross-sectional study in patients with mild hemiplegia

Daigo Sakamoto[1,2], Toyohiro Hamaguchi[2]*, Naohiko Kanemura[2], Takashi Yasojima[2], Keisuke Kubota[3], Ryota Suwabe[1], Yasuhide Nakayama[4], Masahiro Abo[4]*

1 Department of Rehabilitation Medicine, The Jikei University School of Medicine Hospital, Tokyo, Japan, 2 Department of Rehabilitation, Graduate School of Health Science, Saitama Prefectural University, Saitama, Japan, 3 Research Development Center, Saitama Prefectural University, Saitama, Japan, 4 Department of Rehabilitation Medicine, The Jikei University School of Medicine, Tokyo, Japan

* hamaguchi-toyohiro@spu.ac.jp (TH); abo@jikei.ac.jp (MA)

## Abstract

The reaching motion to the back of the head with the hand is an important movement for daily living. The scores of upper limb function tests used in clinical practice alone are difficult to use as a reference when planning exercises for movement improvements. This cross-sectional study aimed to clarify in patients with mild hemiplegia the kinematic characteristics of paralyzed and non-paralyzed upper limbs reaching the occiput. Ten patients with post-stroke hemiplegia who attended the Department of Rehabilitation Medicine of the Jikei University Hospital and met the eligibility criteria were included. Reaching motion to the back of the head by the participants' paralyzed and non-paralyzed upper limbs was measured using three-dimensional motion analysis, and the motor time, joint angles, and angular velocities were calculated. Repeated measures multivariate analysis of covariance was performed on these data. After confirming the fit to the binomial logistic regression model, the cutoff values were calculated using receiver operating characteristic curves. Pattern identification using random forest clustering was performed to analyze the pattern of motor time and joint angles. The cutoff values for the movement until the hand reached the back of the head were 1.6 s for the motor time, 55˚ for the maximum shoulder joint flexion angle, and 145˚ for the maximum elbow joint flexion angle. The cutoff values for the movement from the back of the head to the hand being returned to its original position were 1.6 s for the motor time, 145˚ for the maximum elbow joint flexion angle, 53˚/s for the maximum angular velocity of shoulder joint abduction, and 62˚/s for the maximum angular velocity of elbow joint flexion. The numbers of clusters were three, four, and four for the outward non-paralyzed side, outward and return paralyzed side, and return non-paralyzed side, respectively. The findings obtained by this study can be used for practice planning in patients with mild hemiplegia who aim to improve the reaching motion to the occiput.

**Data Availability Statement:** Uploading this study's dataset to public repositories is restricted by the ethics committee that approved this study.

This data contain potentially identifying or sensitive patient information; therefore, researchers who wish to use this dataset may obtain it after submitting an individual application to the following ethics committee: The Jikei University Institutional Review Board, 3-25-8 Nishi-Shimbashi, Minato-ku, Tokyo, 105-8461 Tel: +81-3-3433-1111 Ext 2187 Fax: +81-3-5400-1388    Email: rinri@jikei.ac.jp.

**Funding:** The author(s) received no specific funding for this work.

**Competing interests:** The authors have declared that no competing interests exist.

## Introduction

Stroke affects 14 million individuals annually worldwide, and 80 million people live with the aftereffects of stroke [1]. Motor paralysis, a sequela of stroke, occurs in approximately 80% of patients [2]. As motor paralysis of the upper limbs and fingers limits patients' activities of daily living (ADLs) and decreases their quality of life (QOL), patients are provided with continuous rehabilitation to improve their motor capacities [3, 4].

Reaching the occiput is an important exercise for achieving rehabilitation. Reaching is a basic movement of the upper extremity aimed at extending the arm toward a designated target in order to touch or grasp something [5]. Although there are various target points for reaching, such as a space or an object on a desk, a reach whose target point is one's body directly influences self-care performance. The movement of the hand to the back of the head is included in movements for grooming, such as washing, tying hair, and putting on and taking off clothes, ornaments, and hats [6]. Improving the patient's appearance positively affects their QOL through social participation, such as going out and socializing. Reaching the back of the head requires a wide range of motion and coordination between the shoulder and elbow joints, which is challenging for patients with motor paralysis. Besides the occiput, the mouth, chest, abdomen, and lower back are other reaching target points for eating, dressing, and bathing. While important for self-care performance, reaching these target points is less difficult than reaching the occipital area; thus, patients with mild hemiplegia can often perform these reaches adequately [7, 8]. The ability to smoothly reach the back of the head, which is difficult for hemiplegics, is beneficial to their ADLs. Even if the patient's motor paralysis is mild, smooth movement requires practice [9, 10]. Therefore, therapists monitor patient changes owing to exercise and treatment and provide further treatment based on the evaluation.

Clinically, the arm function test, manual function test, and action research arm test (ARAT) are used for upper extremity function assessment to observe the reach to the occiput [8, 11]. In these tests, the examiner observes whether the patient can perform the task and the reaching position of the patient's hand at the end of the motor limb; the test is scored using an ordinal scale. However, as the human body has redundant degrees of freedom, multiple combinations of joint motions exist, even when the final hand position is the same [12]. Specifically, synergistic patterns may emerge in patients with hemiplegia after stroke, and compensatory movements may be used [13]. In patients with motor paralysis, even if reaching the occiput is possible, the motion trajectory may be prolonged, and the motion time may be delayed. Due to a ceiling effect, upper extremity function scores used in clinical practice do not measure the recovery in patients with mild motor paralysis [14, 15]. When planning a change in joint movement pattern exercises or to shorten their duration, attempting to use the clinically obtained scores is challenging. To the best of our knowledge, a kinematic index that quantitatively expresses the "reach to the back of the head" does not exist. Moreover, a clear target value to determine if the reach has been sufficiently recovered in patients with mild motor paralysis is lacking.

Three-dimensional motion analysis (3DMA) is used to analyze upper limb movement characteristics in patients who have suffered stroke [16]. Studies have been conducted to analyze the joint motion and motion time of patients with stroke using 3DMA [17, 18]. Kinematic quantification of movement is strongly recommended to distinguish between the recovery of motor strategies and improvement owing to motor compensations; therefore, it is increasingly used in clinical practice as a new outcome measure [19, 20]. Kinematic characteristics calculated by analyzing movements related to daily life are strongly related to the patient's activity capacity and are valid indicators of patient treatment in clinical practice [21–24]. Several studies have evaluated healthy participants and patients with orthopedic diseases regarding their

ability to reach the occipital region; however, kinematic characteristics such as motor time, angular change of joints, and angular velocity of joints have not been validated in stroke patients with mild motor paralysis [9, 10, 25]. This issue can be analyzed using 3DMA to provide quantified data. The data on the kinematic characteristics of patients with stroke with regard to reaching backward to the occiput would provide a useful basis for devising a practice method for those who aim to acquire ADLs, including reaching backward to the occiput.

Upper extremity motor tasks can be analyzed using 3DMA by comparing healthy subjects and patients; however, in clinical situations, movements on the paralyzed and non-paralyzed side of the patient may be compared [26]. The movements of the non-paralytic upper limb in patients with hemiplegia after a stroke differ from those of normal participants, and the reference for a paralyzed movement is not necessarily the corresponding movement of the non-paralyzed limb. Although the non-paralyzed upper limb may also have compensatory strategies for movement, a comparison of the kinematic characteristics of the paralyzed and non-paralyzed upper limbs of patients with hemiplegia with mild impairment of limb and trunk function has been conducted to validate the calculation of the evaluation values that will be used as a reference for treatment [27, 28]. 3DMA is clinically useful in patients with mild hemiparesis because the motion of the non-paralyzed upper limb is referenced, and the joint motion patterns of the paralyzed upper limb can be used to devise isolation exercises to be practiced [22]. In addition, clustering patterns of movement time and joint angle changes from the data obtained using opportunity learning would be useful for providing appropriate exercises for patients [29].

Based on the above discussion, this study aimed to clarify the kinematic characteristics of paralyzed and non-paralyzed upper limbs in patients with chronic phase mild hemiplegia while reaching the occipital region with the hand using 3DMA. The cutoff values calculated from the measured data and the clustered patterns of the movement time and joint angle changes can be used as target values in clinical practice and to plan clinical practice. Our results will provide a basis for devising motor targets and effective exercises to reach the occiput in patients with posterior upper limb hemiplegia.

## Materials and methods

### Study design

In this cross-sectional study, kinematic data were obtained from patients with post-stroke hemiplegia while reaching the back of their head with their hand, and the patient's paralyzed and non-paralyzed upper limbs were compared.

### Ethical considerations

All patients provided written informed consent to participate in this study. This study was approved by the Ethics Committee of the Jikei University School of Medicine (approval number: 22-061-6238).

### Participants

We included outpatients aged ≥20 years with a history of at least 6 months of post-stroke hemiplegia who attended the Department of Rehabilitation Medicine of the Jikei University Hospital between May 1 and October 30, 2020. The patients presented with a total score of ≥47 points in the Fugl–Meyer assessment of the upper extremity (FMA-UE) or a gross movement score of ≥6 points in the ARAT. The 47 points achieved in the FMA-UE is the cutoff for mild motor paralysis in the severity classification of motor paralysis reported by Woodbury

et al. [30]. The gross movement in the ARAT includes reaching the back of the head, and 6 points of gross movement is the threshold for patients who can perform this movement [8]. The non-inclusion criteria were as follows: cases with a paralyzed hand not reaching the external occipital ridge with automatic movements; presence of a central nervous system disease other than stroke, orthopedic disease, mental disorder, cognitive impairment after stroke, dementia, visual field disorder, and ataxia at diagnosis; subluxation of the shoulder joint; pain in the joints of the upper limb or fingers during exercise; presence of a significant limitation in the joint range of motion in the upper limb; and completion of the occupational therapy intervention. Patients who met the study eligibility criteria but did not meet the non-inclusion criteria were asked to participate in the study, and those who provided consent were considered participants.

## Sample size

Using G*Power 3.1 (University of Düsseldorf, Düsseldorf, Germany), the minimum sample size was calculated to be eight patients. To calculate the cutoff values of the kinematic data by performing binomial logistic regression analysis using the paralyzed and non-paralyzed upper limbs as nominal variables in patients with hemiplegia, data from previous studies [31], in which the primary assessment was the joint angle, were referenced. The sample size was calculated by setting the difference from the constant (test family: exact test, binomial test, one sample case). For calculating the required sample size, the effect size (g) was 0.4, α was 0.05, power (1-β) was 0.8, and the constant proportion was 0.5.

## Survey period

The acquisition of patients' medical information, clinical evaluation, and measurement of motor tasks began on October 1, 2020, and ended on October 1, 2021.

## Experimental procedure

The examiner made the participants sit on a backless chair. The starting position of the upper limb on the measurement side was set with the elbow in full extension, fingers in extension, and the forearm in mid-position between supination and pronation; the forearm and fingers were not in contact with the chair. The upper limb on the non-tested side was also placed in the same position (Fig 1). If the starting position was difficult to achieve owing to paralysis and muscle tone, the participants were reminded to relax the muscles of the upper limbs and fingers, and the upper limbs were allowed to droop as much as possible. The examiner adjusted the height and position of the chair so that the participants' forearms and fingers did not touch the chair, and the flexion angles of the knee and hip joints were 90˚. The participants sat on a height-adjusted chair with their feet shoulder-width apart and both soles of their feet on the floor. Six infrared cameras were placed on the room's ceiling, and a landmark was placed 5 m away from the participants (Fig 2). The landmark was placed to minimize, by looking at it, the neck and trunk movement of participants performing the reaching task.

The examiner affixed infrared reflective markers at 35 points on the participant's body according to the Plug-in Gait marker model (Fig 3) [32]. Infrared reflective markers were applied by a single examiner to all the participants following a similar procedure. On the head, they were placed at four bilateral anterolateral and posterolateral points; on the trunk, they were placed at five points on the upper sternal body, lower sternal spine, seventh cervical vertebra, 10th thoracic vertebra, and proper posterior back; and on the pelvis, at four points on the bilateral anterior and posterior superior iliac spines. On the upper limbs, they were affixed at 10 points on the bilateral upper acromion, lateral olecranon, radial and ulnar eminence, and

a
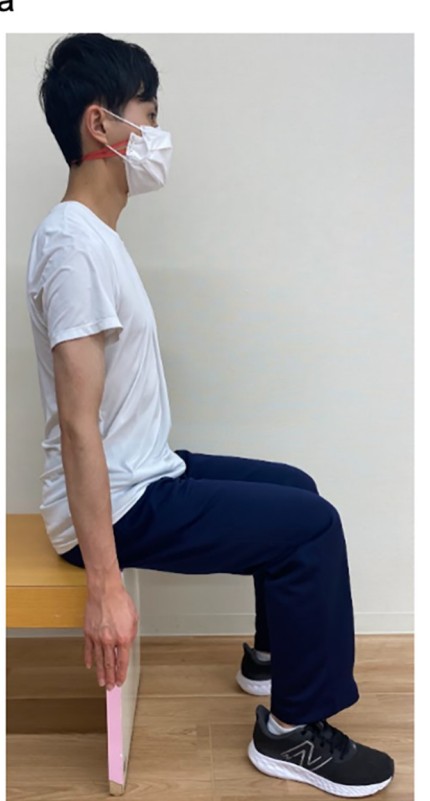

b
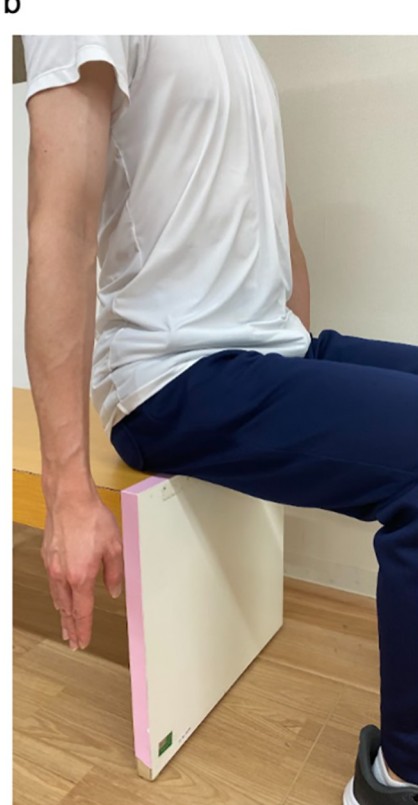

**Fig 1. Starting position during measurement.** (a) Side view of the starting position and (b) the starting position of the upper limb.

the head of the second metacarpal. On the lower limbs, they were affixed at 12 points on the bilateral femur, lateral patella, tibia, calcaneus, external calcaneus, and head of the second metatarsal bone.

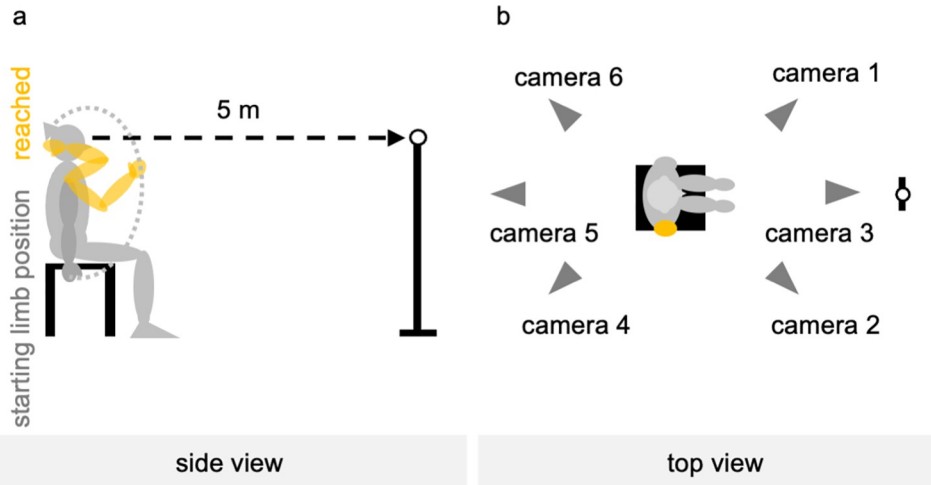

**Fig 2. Measurement environment.** (a) Side view and (b) top view. A landmark was placed 5 m away from the participants. By looking at it, neck and trunk movement was minimized in participants performing the reaching task.

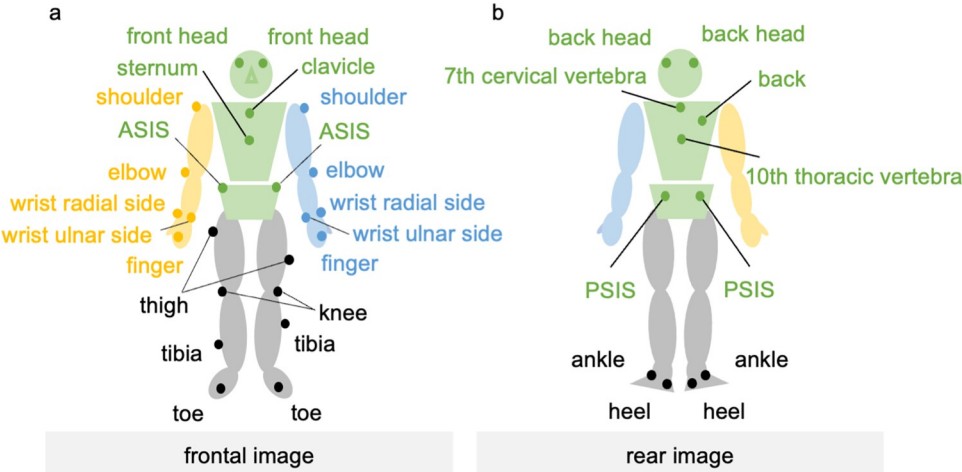

**Fig 3. Attachment positions of the markers.** (a) Frontal image and (b) rear image. The circled positions indicate infrared markers, which were attached to 35 points on the participants' bodies to construct the Plug-in Gait model.

## Reaching task

The reaching task involved placing the palmar aspect of the hand (the palmar aspect of the second metacarpal) in contact with the center of the external occipital ridge. The examiner provided the following verbal instructions to all participants: "touch the center of the occiput with the palm of the hand"; "return the upper limbs to the starting posture"; "maintain the head, neck, and trunk as still as possible while looking at the landmarks placed in front during the measurement"; and "move the arms as usual without any particular awareness of speed." Flexion of the fingers during the reaching motion was allowed. The examiner asked the participants to perform the exercise several times and checked whether they understood the instructions correctly.

Measurements were performed on the paralyzed and non-paralyzed sides, in that order, five times each, for a total of 10 measurements (Fig 4). A 1-minute rest period was allowed between the measurements. Before each measurement, the examiner instructed the participants to attempt to achieve the starting position with the upper limb as much as possible. If a participant was unable to assume the starting position due to heightened muscle tone in their upper limb, causing a slight bending of the elbow joint, the procedure was still considered appropriate. The order of the measurements did not change, and all participants underwent

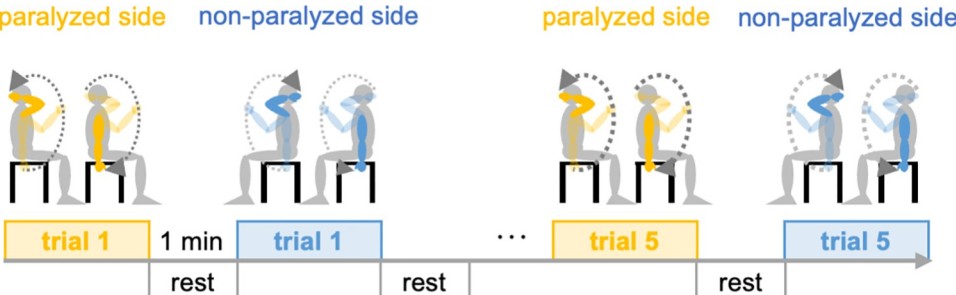

**Fig 4. Measurement sequence.** The illustrations on the left of each trial show the outward motion from the starting position until the hand reaches the back of the head, and the illustrations on the right show the return motion from the point where the hand reaches the back of the head to the original position.

measurements in the same manner. During the rest period, the participants were allowed to stretch the muscles by themselves but were not allowed to receive any therapeutic intervention from the therapist.

## Data acquisition and analysis

The reaching task was analyzed using an optical 3DMA (Vicon motion system; Oxford, UK). Data were recorded at a sampling rate of 100 Hz using six infrared cameras mounted on the ceiling of the examination room. The displacement information for each marker was compiled in three dimensions using parallax images from the six cameras, converted into positions (x, y, and z) in the same virtual space, and recorded on a personal computer for analysis. The motion was divided into an outward motion from the starting position until the hand reached the occiput (outward motion) and a return motion from the point when the hand reached the occiput until it returned to its original position (return motion). The onset of the outward and return motions was defined as the time when the position data of the index finger marker in the three-dimensional space changed continuously for 0.2 s. The end of the movement was defined as the time when the position data of the index finger marker recorded the same value continuously for 0.2 s [33].

A whole-body rigid-body linkage model was constructed from the acquired reflective marker information using the Plug-in Gait model specified by the Vicon motion system [32]. The flexion and abduction angles of the shoulder joint, as well as the flexion angle of the elbow joint, were calculated using the Eulerian method, in which the joint angles are calculated from the distal body segment coordinates in motion relative to the proximal body segment coordinates. The angular velocity was calculated by dividing the change in the joint angle by the time of motion. The displacements of the markers attached to the sternal pattern and occiput on the same side as the measured upper limb during the measurement were calculated using the position data in three-dimensional space. First, the peak values were extracted from data obtained from 10 participants, each with five measurements on one upper limb (50 data in total). Next, the mean, standard deviation, maximum, and minimum values of the joint angles, angular velocities of shoulder flexion, shoulder abduction, and elbow flexion, and displacements of the sternal pattern and occipital markers were calculated separately for outward and return motions. Data from the same participant were not averaged to allow subsequent analyses using random forest clustering.

## Clinical evaluation

The FMA was used to assess the motor paralysis of the participants [34]. Upper (FMA-UE) and lower (FMA-LE) extremity motor function items can be used in excerpts and assess isolated movements in accordance with the recovery phase of motor paralysis. The FMA-UE and FMA-LE are scored on 66- and 34-point scales, respectively, based on a 3-point ordinal scale. The ARAT and Box and Block Test (BBT) were used to assess the participant's ability to manipulate objects. The ARAT is an upper limb function assessment tool, which was developed based on the upper extremity test [35]. The ARAT consists of grasp, grip, pinch, and gross movement subtests and is scored on a 57-point scale based on a 4-point ordinal scale [8]. The BBT is used to evaluate hand dexterity. The task is to move 100 blocks from one compartment of a box to the opposite compartment one by one across a partition [36, 37]. In this test, the number of blocks moved per minute is measured. The modified Ashworth scale (mAs) was used to assess the muscle tone of the study participants. The mAs is used to evaluate spasticity, a symptom of abnormal muscle tone in central nervous system diseases [38]. In this test, the resistance to rapid movement of the joints in an alternating manner is evaluated on a

6-point scale. In this study, the biceps brachii muscle of the paralyzed side of the participants was evaluated. The Berg balance scale (BBS) and the functional reach test (FRT) were used to assess the participant's balance function. The BBS is used to evaluate functional balance ability and is useful as an indicator of walking independence and fall prediction [39, 40]. The test is scored on a 5-point ordinal scale on a 56-point scale. The FRT is a balance ability assessment that measures the distance of the forward reach of the upper limb in the standing position without changing the basal plane of support [41]. The FRT was performed three times in this work, and the mean value was calculated. The Semmes–Weinstein monofilament test (SWT) and the thumb search test (TST) were used to evaluate the sensory function of the participants. The SWT examines static tactile sensations related to object properties, discrimination ability, and sustained grasping [42]. In this test, five types of nylon filaments with different diameters are used to stimulate the skin, and the participant's responses are evaluated. The TST is used to evaluate the joint localization of the upper limb [43]. This test integrates the proprioceptive information of each upper limb joint and evaluates the perception of thumb position in space on a 4-point ordinal scale. The Barthel Index (BI) was used to assess participants' daily functioning in this study. The BI is used to assess the level of independence in performing ADLs [44]. The test consists of 10 items, and the degree of assistance is evaluated using a 4-point ordinal scale. If the patient independently performs all items, the score is 100 points; if the patient requires full assistance for all items, the score is 0 points.

## Participant characteristics

Basic and medical information regarding the participants, including sex, age, height, body mass index (BMI), stroke type, post-stroke duration, and paralytic side, was collected from their medical records. The Edinburgh Handedness Test was used to examine the handedness of the participants [45]. This test consists of 10 questions, in which the participants are asked which hand they use to perform ADLs. A positive index indicates right-handedness, whereas a negative index indicates left-handedness. In this study, the dominant hand before the stroke and the current dominant hand were investigated.

## Statistical analysis

Assuming that there is no left-right difference in the movement pattern between the dominant and non-dominant upper limbs for reaching the back of the head with the hand in normal individuals without motor paralysis, we hypothesized that movement time, joint angles, and angular velocity while performing the movement of reaching the back of the head with the hand in patients with chronic phase mild hemiplegia will differ between the paralyzed and non-paralyzed upper limbs. A repeated measures multivariate analysis of covariance (within-subject design) was performed to test this hypothesis. The dependent variables were motor time, maximum values of joint angles, and angular velocities of shoulder flexion, shoulder abduction, and elbow flexion, and the fixed factor was the measured side of the reaching task. The covariates included sex, age, BMI, time since stroke onset, maximum displacement of the sternal pattern, and occipital markers. The parameter $\eta^2$ was used to determine the effect size in statistical analyses, and the effect size indices were set as 0.01 for small, 0.06 for medium, and 0.14 for large [46]. Variables for which the Shapiro–Wilk test confirmed that the data were not normally distributed were compared using the Kruskal–Wallis test. Statistical analyses were conducted separately for the outbound and return motions. Binomial logistic regression analysis was conducted on the features fitted to the model for the paralyzed and non-paralyzed sides using repeated measures multivariate analysis of covariance. After confirming the fit of the binomial logistic regression model, Youden's index was calculated using the receiver

operating characteristic (ROC) curve, and the cutoff value to discriminate between the paralyzed and non-paralyzed sides was calculated. The calculated cutoffs, especially the area under the curve (AUC) values, were compared using the Delong test [47]. The dependent variables were the paralyzed and non-paralyzed sides, and the independent variables were the characteristics that showed significant differences between the paralyzed and non-paralyzed sides in the multivariate analysis. The covariates included sex, age, BMI, post-onset period, and the maximum displacement values of sternal pattern markers and occipital markers. For these analyses, jamovi version 2.2.1 (https://www.jamovi.org) was used.

Pattern identification using random forest clustering was performed to analyze the pattern of changes in the joint angle on the paralyzed and non-paralyzed sides of reaching the occipital region. Random forest clustering is an algorithm that divides the data into several clusters, virtually partitioning the data such that each observation belongs to only one group. The clustering method is an unsupervised method that uses the random forest algorithm, wherein the dependent variable $y$ is set as $T$, the number of decision trees, and the machine is trained with each motor data input as unlabeled data with and without motor paralysis (Eq 1). The random forest algorithm generates a proximity matrix that estimates the distance between the observations based on the frequency of observations ending at the same leaf node (Eq 2). These data consist of continuous variables [48–50]. To determine which features to use as data partition nodes in Eq 1, information gain was measured using a random forest in Eq 2. The information gain is calculated for all the possible split points (each feature's value) in the dataset, and the feature f and its split point with the largest information gain were selected. Random forests are characterized by the random selection of a subset of features to be used in the construction of each decision tree. The process of selecting the feature f that maximizes the information gain in Eq 2 from the features in Eq 1 data ensures model diversity, suppresses overlearning, and improves generalization performance.

$$(x, y) = (x1, x2, x3, \ldots, xk, y) \tag{Eq1}$$

$$IG\left(D_p, f\right) = I\left(D_p\right) - \frac{N_{non-paretic}}{N_p} I\left(D_{non-paretic}\right) - \frac{N_{paretic}}{N_p} I\left(D_{paretic}\right) \tag{Eq2}$$

In Eq 1, $x$ is the feature value of the paralyzed side, $y$ is the value of the paralyzed side as the objective variable, and $k$ is the total number of features. In Eq 2, $IG$ is the information gain in a random forest, which is the sum of the impurity of the left and right child nodes from the impurity of the parent node, weighted by their ratio; $Dp$ is the parent dataset; $f$ is the number of explanatory variables; $N$ is the depth of the decision tree; $D_{non-paretic}$ is the non-paretic child node from the parent node; $D_{paretic}$ is the paretic child node from the parent node; and $I$ is the impurity. The variables used were the motor time and the shoulder flexion, shoulder abduction, and elbow flexion angles. Each variable $C$ was assigned a discrete-valued class label, that is, C0, C1, C2,..., Cn, using random forest clustering. The motor patterns were clustered for four conditions: outward and return motions on the paralyzed and non-paralyzed sides of the reach to the occipital region. The number of clusters was determined using the elbow method. The elbow method plots the sum of squares of the intra-cluster error for each cluster and considers the point where the value sharply decreases to be the optimal number of clusters [51]. The clustering structure of these four conditions is illustrated and is considered a unique pattern underlying the motion data. JASP version 0.16 (https://jasp-stats.org/) was used to identify these patterns. The statistical significance level was set at 5%.

## Results

### Participants

In total, 88 patients with chronic stroke had a history of intervention in the previous 6 months. Among these patients, three were aged <20 years, and 60 had FMA-UE or ARAT scores below the inclusion criteria. One patient was diagnosed with cognitive impairment after stroke, two had ataxia, two had shoulder subluxation, and seven completed the occupational therapy intervention. Of the 88 patients, 75 were excluded, and 13 met the inclusion criteria set for the study. These 13 patients were asked to participate in the study; three patients who did not agree to participate were excluded; therefore, 10 patients were included in the final analysis (Fig 5).

The characteristics of the participants and the survey results are presented in Table 1. In total, 10 participants were included, of whom three were female and seven were male individuals. All the participants had right hemiplegia. The right hand was dominant in 10 participants before the onset of the stroke, and the dominant side switched to the left hand in 5 out of 10 participants after the onset. The severity of FMA-UE was moderate in one patient and mild in nine patients. One participant classified as moderate met the eligibility criterion of at least 6 points for gross movement on the ARAT. The median scores of all the participants were 34 points on the ARAT and 24 points on the paralytic side in the BBT. Eight and two patients had mAs scores of 1 and 1+, respectively. None of the patients showed severe spasticity. The FMA-LE score, BBS score, and FRT were 28 points, 55 points, and 33 cm, respectively. The participants had good lower limb, trunk, and balance functions. The SWT was normal in seven participants and lowered in three, whereas the TST was normal in nine participants and lowered in one; there was no significant decrease in the superficial or deep sensation in the participants. All participants had a BI score of 100 points and independently performed all daily activities.

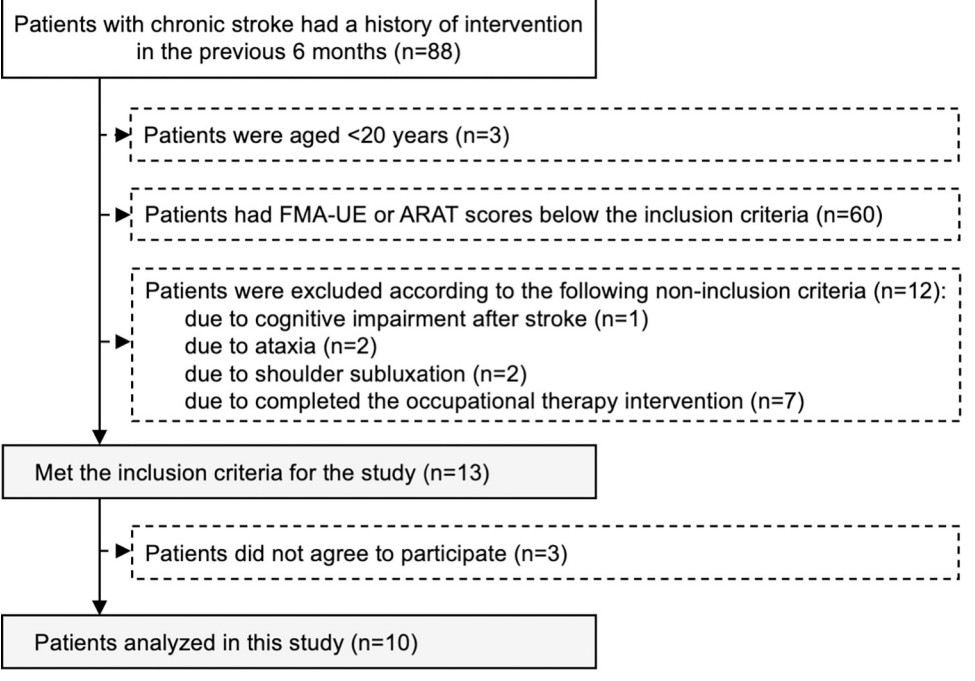

**Fig 5. Patient selection procedure.**

**Table 1. Clinical characteristics of the patients.**

| Characteristics | | Female | Male | All |
|---|---|---|---|---|
| Participants | | 3 (30) | 7 (70) | 10 (100) |
| Age (years) | | 50 [35–51] | 46 [46–55] | 48 [45–51] |
| Height (cm) | | 154 [152–159] | 170 [169–173] | 169 [164–171] |
| Weight (kg) | | 51 [51–56] | 72 [66–77] | 66 [55–73] |
| BMI (kg/m$^2$) | | 23 [21–24] | 25 [22–26] | 25 [20–25] |
| Time from onset (months) | | 59 [37–83] | 53 [38–78] | 54 [34–91] |
| Diagnosis | CI | 3 (100) | 3 (43) | 6 (60) |
| | ICH | 0 (0) | 4 (57) | 4 (40) |
| Paralyzed side | Left | 0 (0) | 0 (0) | 0 (0) |
| | Right | 3 (100) | 7 (100) | 10 (100) |
| Dominant hand | | | | |
| Before onset | Left | 0 (0) | 0 (0) | 0 (0) |
| | Right | 3 (100) | 7 (100) | 10 (100) |
| After onset | Left | 2 (67) | 3 (43) | 5 (50) |
| | Right | 1 (33) | 4 (57) | 5 (50) |
| FMA-UE | | 56 [52–57] | 57 [55–60] | 57 [55–58] |
| FMA-UE severity | | | | |
| Moderate (20≤ score ≤46) | | 0 (0) | 1 (14) | 1 (10) |
| Mild (score ≥47) | | 3 (100) | 6 (86) | 9 (90) |
| FMA-LE | | 28 [25–28] | 30 [28–31] | 28 [28–30] |
| ARAT | | 31 [23–34] | 43 [28–50] | 34 [27–45] |
| BBT | Paralyzed side | 22 [12–25] | 25 [10–38] | 24 [7–28] |
| | Non-paralyzed side | 53 [53–56] | 59 [53–61] | 57 [53–60] |
| mAs | 0 | 0 (0) | 0 (0) | 0 (0) |
| | 1 | 2 (67) | 6 (86) | 8 (80) |
| | 1+ | 1 (33) | 1 (14) | 2 (20) |
| BBS | | 55 [54–55] | 56 [55–56] | 55 [55–56] |
| FRT (cm) | | 29 [28–30] | 38 [33–40] | 33 [29–39] |
| SWT | Normal | 3 (100) | 4 (57) | 7 (70) |
| | Decline | 0 (0) | 3 (43) | 3 (30) |
| TST | Normal | 3 (100) | 6 (86) | 9 (90) |
| | Decline | 0 (0) | 1 (14) | 1 (10) |
| BI | | 100 [100] | 100 [100] | 100 [100] |

Values are expressed as numbers (%) or medians [25$^{th}$–75$^{th}$ percentile].

ARAT, action research arm test; BBS, Berg balance scale; BBT, Box and Block Test; BI, Barthel Index; BMI, body mass index; CI, cerebral infarction; FMA-LE, Fugl–Meyer assessment of the lower extremity; FMA-UE, Fugl–Meyer assessment of the upper extremity; FRT, functional reach test; ICH, intracranial hemorrhage; mAs, modified Ashworth scale; SWT, Semmes–Weinstein monofilament test; TST, thumb search test.

## Detection of paralyzed upper limb with motion data

Tables 2 and 3 present the results for the kinematic data on outward and inward reaching motions. Fig 6 shows the changes in the joint angles of shoulder flexion, shoulder abduction, and elbow flexion during the outward and return motions of the reaching motion.

Repeated measures multivariate analysis of covariance was performed. For the outward motion of the reaching task, Wilks' lambda test showed a significant main effect (F [7, 86] = 25.1, $p < 0.001$) on the measurement side. The paralyzed side had a significantly longer motor time (F = 136.6, $p < 0.001$, $\eta^2 = 0.51$). The peak shoulder flexion angle was significantly greater

**Table 2. Outward motion of the reaching task: Results of multivariate analysis of covariance.**

| Descriptive statistics | | Paralyzed side | | | | Non-paralyzed side | | | | F | p | $\eta^2$ |
|---|---|---|---|---|---|---|---|---|---|---|---|---|
| | | Mean | SD | Min | Max | Mean | SD | Min | Max | | | |
| Motor time (s) | | 2.3 | 0.7 | 1.5 | 4.4 | 1.3 | 0.3 | 0.9 | 1.9 | 136.6 | <0.001 | 0.51 |
| Peak angle (°) | Shoulder flex | 48.9 | 15.4 | 24.4 | 75.1 | 42.3 | 11.2 | 23.2 | 59.5 | 15.3 | <0.001 | 0.07 |
| | Shoulder abd | 117.5 | 16.7 | 88.8 | 156.8 | 118.9 | 6.2 | 102.7 | 129.2 | 0.3 | 0.560 | 0.01 |
| | Elbow flex | 134.2 | 8.0 | 113.0 | 145.0 | 140.0 | 6.1 | 120.6 | 145.6 | 19.5 | <0.001 | 0.16 |
| Peak angular velocity (°/s) | Shoulder flex | 213.6 | 190.8 | 71.8 | 893.6 | 180.5 | 88.5 | 70.6 | 524.8 | 1.4 | 0.237 | 0.00 |
| | Shoulder abd | 306.0 | 249.4 | 90.3 | 1116.3 | 271.3 | 109.6 | 142.6 | 792.5 | 0.9 | 0.339 | 0.00 |
| | Elbow flex | 211.6 | 118.6 | 80.6 | 658.6 | 252.9 | 142.5 | 115.3 | 887.0 | 2.9 | 0.090 | 0.03 |
| Displacement (mm) | Manubrium | 4.8 | 4.7 | 0.5 | 16.4 | 2.9 | 4.4 | 0.4 | 29.3 | - | - | - |
| | Back of the head | 1.0 | 19.9 | 1.8 | 141.2 | 8.5 | 6.0 | 0.5 | 24.2 | - | - | - |

abd, abduction; flex, flexion; Max, maximum; Min, minimum; SD, standard deviation.

Multivariate analysis of covariance was used, with the statistical significance set at 0.05 (N = 10). The manubrium and back of the head displacements were used as covariates.

on the paralyzed side (F = 15.3, $p < 0.001$, $\eta^2 = 0.07$), and the peak elbow flexion angle was significantly greater on the non-paralyzed side (F = 19.5, $p < 0.001$, $\eta^2 = 0.16$). There were no differences in the peak shoulder abduction angle (F = 0.3, $p = 0.560$, $\eta^2 = 0.07$) and peak angular velocities of shoulder flexion (F = 1.4, $p = 0.237$, $\eta^2 = 0.00$), shoulder abduction (F = 0.9, $p = 0.339$, $\eta^2 = 0.00$), and elbow flexion (F = 2.9, $p = 0.090$, $\eta^2 = 0.03$; Table 2). For the return motion of the reaching task, Wilks' lambda test showed a significant main effect (F [7, 86] = 31.5, $p < 0.001$) on the measured side. Motor time (F = 98.4, $p < 0.001$, $\eta^2 = 0.43$) was significantly longer on the paralyzed side. The peak angular velocities of shoulder flexion (F = 7.4, $p = 0.008$, $\eta^2 = 0.07$), shoulder abduction (F = 13.9, $p < 0.001$, $\eta^2 = 0.12$), and elbow flexion (F = 9.4, $p = 0.003$, $\eta^2 = 0.09$) were significantly greater on the paralyzed side. The peak elbow flexion angle was significantly greater on the non-paralyzed side (F = 19.0, $p < 0.001$, $\eta^2 = 0.15$). There was no significant difference in the peak shoulder flexion angle (F = 3.7, $p = 0.057$, $\eta^2 = 0.02$) or peak shoulder abduction angle (F = 3.2, $p = 0.075$, $\eta^2 = 0.03$; Table 3).

**Table 3. Return motion of the reaching task: Results of multivariate analysis of covariance.**

| Descriptive statistics | | Paralyzed side | | | | Non-paralyzed side | | | | F | p | $\eta^2$ |
|---|---|---|---|---|---|---|---|---|---|---|---|---|
| | | Mean | SD | Min | Max | Mean | SD | Min | Max | | | |
| Motor time (s) | | 2.3 | 0.7 | 1.5 | 4.2 | 1.5 | 0.3 | 1.0 | 2.2 | 98.4 | <0.001 | 0.43 |
| Peak angle (°) | Shoulder flex | 50.3 | 15.1 | 25.2 | 70.2 | 46.9 | 12.8 | 23.9 | 71.9 | 3.7 | 0.057 | 0.02 |
| | Shoulder abd | 116.9 | 16.2 | 89.2 | 156.6 | 121.1 | 6.9 | 102.7 | 139.0 | 3.2 | 0.075 | 0.03 |
| | Elbow flex | 134.2 | 8.1 | 110.3 | 145.1 | 140.0 | 6.1 | 120.5 | 145.0 | 19.0 | <0.001 | 0.15 |
| Peak angular velocity (°/s) | Shoulder flex | 122.7 | 103.4 | 13.1 | 405.6 | 79.2 | 43.9 | 17.6 | 235.6 | 7.4 | 0.008 | 0.07 |
| | Shoulder abd | 81.7 | 87.1 | -4.3 | 359.7 | 34.2 | 32.3 | -5.6 | 132.9 | 13.9 | <0.001 | 0.12 |
| | Elbow flex | 95.4 | 98.1 | -1.0 | 428.5 | 48.9 | 41.8 | -12.1 | 187.5 | 9.4 | 0.003 | 0.09 |
| Displacement (mm) | Manubrium | 29.6 | 167.6 | 0.5 | 1190.0 | 4.1 | 4.9 | 0.5 | 19.6 | - | - | - |
| | Back of the head | 9.1 | 20.2 | 1.4 | 145.4 | 5.9 | 4.4 | 0.7 | 19.4 | - | - | - |

abd, abduction; flex, flexion; Max, maximum; Min, minimum; SD, standard deviation.

Multivariate analysis of covariance was used, with the statistical significance set at 0.05 (N = 10). The displacements of the manubrium and back of the head were used as covariates.

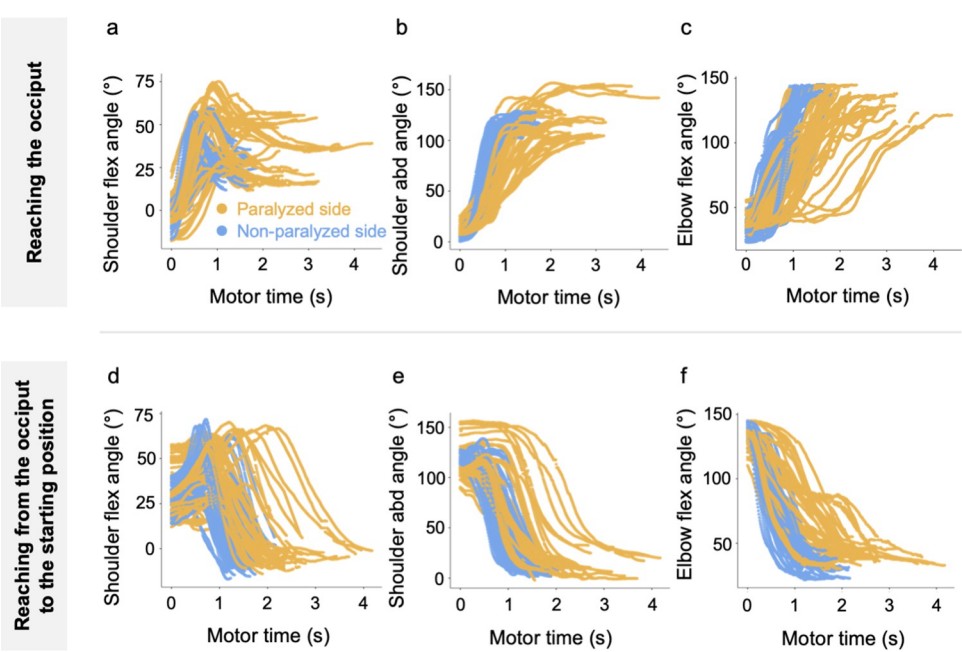

**Fig 6. Changes in the joint angle during the reaching motion.** (a) Changes in the shoulder flex angle, (b) changes in the shoulder abd angle, and (c) changes in the elbow flex angle while reaching the occiput and (d) changes in the shoulder flex angle, (e) changes in the shoulder abd angle, and (f) changes in the elbow flex angle while reaching from the occiput to the starting limb position. abd, abduction; flex, flexion.

Binomial logistic regression analysis of the model-fitted features on the paralyzed and non-paralyzed sides was performed using repeated measures multivariate analysis of covariance. During the outward motion, the regression model was fitted for the motor time ($X^2 = 112$, $p < 0.001$), peak shoulder flexion angle ($X^2 = 18.4$, $p = 0.010$), and peak elbow flexion angle ($X^2 = 24.9$, $p < 0.001$; Table 4 and Fig 7). During the return motion, the regression model was fitted for the motor time ($X^2 = 87.5$, $p < 0.001$), peak elbow flexion angle ($X^2 = 28.8$, $p < 0.001$), peak angular velocity of shoulder abduction ($X^2 = 18.4$, $p = 0.01$), and peak angular

**Table 4. Model goodness of fit results of the binomial logistic regression analysis.**

| Model | | Deviance | AIC | McFadden's $R^2$ | Overall model test | | | Estimate | 95% CI | | SE | Z | p |
|---|---|---|---|---|---|---|---|---|---|---|---|---|---|
| | | | | | $X^2$ | df | p | | Lower | Upper | | | |
| Reaching motion to the back of the head | | | | | | | | | | | | | |
| Motor time (s) | | 26.4 | 42.4 | 0.8 | 112 | 7 | <0.001 | 19.0 | 6.1 | 32.0 | 14.3 | -2.2 | 0.030 |
| Peak angle (°) | Shoulder flex | 120 | 136 | 0.2 | 18.4 | 7 | 0.010 | 0.01 | 0.04 | 0.2 | 0.03 | 3.1 | 0.001 |
| | Elbow flex | 114 | 130 | 0.2 | 24.9 | 7 | <0.001 | -0.2 | -0.2 | -0.1 | 0,04 | -3.7 | <0.001 |
| Reaching motion from the back of the head to the starting position | | | | | | | | | | | | | |
| Motor time (s) | | 51.1 | 67.1 | 0.6 | 87.5 | 7 | <0.001 | -11.2 | -21.7 | -0.7 | 5.4 | -2.1 | 0.036 |
| Peak angle (°) | Elbow flex | 110 | 126 | 0.2 | 28.8 | 7 | <0.001 | -0.2 | -0.26 | -0.1 | 0.04 | -3.9 | <0.001 |
| Peak angular velocity (°/s) | Shoulder flex | 127 | 143 | 0.1 | 11.8 | 7 | 0.1 | 0.01 | 0.00 | 0.01 | 0.003 | 2.1 | 0.032 |
| | Shoulder abd | 120 | 136 | 0.1 | 18.4 | 7 | 0.01 | 0.01 | 0.004 | 0.02 | 0.01 | 2.9 | 0.004 |
| | Elbow flex | 123 | 139 | 0.1 | 15.8 | 7 | 0.03 | 0.01 | 0.002 | 0.02 | 0.004 | 2.5 | 0.013 |

abd, abduction; AIC, Akaike's information criterion; CI, confidence interval; flex, flexion; SE, standard error.

Binomial logistic regression was used, with the statistical significance set at 0.05 (N = 10).

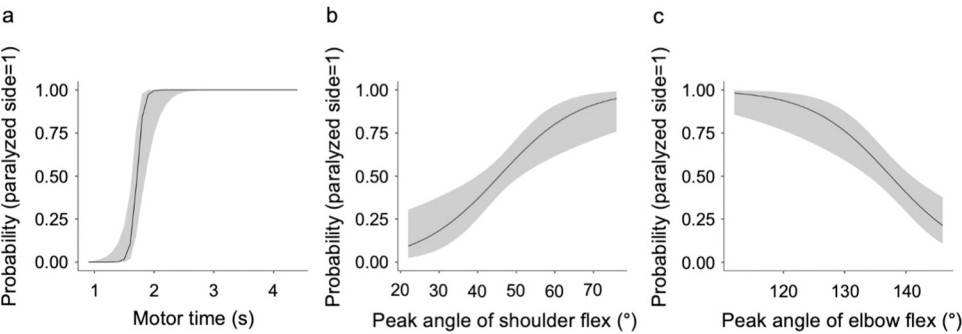

**Fig 7. Results of the binomial logistic regression analysis for the features of reaching motion to the back of the head.** (a) Motor time, (b) peak angle of shoulder flex, and (c) peak angle of elbow flex. flex, flexion. The vertical axis of the figure is probability, and the closer the value is to 1.00, the higher the probability that the upper limb is on the paralyzed side. If the peak shoulder flexion angle was 60° on the outward motion, the probability that the measured upper limb was on the paralyzed side was approximately 75%.

velocity of elbow flexion ($X^2$ = 15.8, $p$ = 0.03). The peak angular velocity of shoulder flexion ($X^2$ = 11.8, $p$ = 0.1) did not fit the regression model (Table 4 and Fig 8).

For features fitted to the regression model, cutoff values were calculated to discriminate between the paralyzed and non-paralyzed sides using the ROC curve. The cutoff values for the outward motion were 1.6 s for the motor time, 55° for the peak shoulder flexion angle, and 145° for the peak elbow flexion angle (Table 5). The cutoff values for the return motion were 1.6 s for the motor time, 145° for the maximum elbow flexion angle, 53°/s for the peak angular velocity of shoulder abduction, and 62°/s for the peak angular velocity of elbow flexion (Table 5). Among the cutoff values detected, the kinematic feature with the highest Youden's index was the motor time for both outward and return motions, whereas the kinematic feature with the largest AUC was the motor time for both the outward motion (AUC = 0.96, SD = 0.02, $p$ < 0.001) and return motion (AUC = 0.92, SD = 0.03, $p$ < 0.001).

## Pattern analysis of upper limb movements

The patterns of changes in the motor time and joint angle on the paralyzed and non-paralyzed sides of the reaching motion to the occipital region were analyzed using random forest clustering. Using the elbow method, the number of clusters was determined from the plotted values of the sum of the squares of the intra-cluster errors for each cluster (Fig 9). The number of clusters was determined to be four for the paralyzed side and three for the non-paralyzed side of the outward motion as well as four for the paralyzed and non-paralyzed sides of the return motion. The clustering structure for the four conditions is presented in Figs 10 and 11. As an example, in the results of the motor time on the paralyzed side of the reaching motion to the occiput, four clusters are illustrated. As the abscissa is the z-value, the motor time of cluster 1 was shorter than the cutoff value. By contrast, cluster 2 had motion times within and longer than the cutoff values, and clusters 3 and 4 had longer motion times than the cutoff values. For example, cluster 4 on the paralyzed side in the outward motion is characterized by longer motor time and greater angular change in shoulder flexion and abduction than the other clusters (Fig 10). For the outward motion, the number of clusters on the paralyzed side was four (N = 11,542; $R^2$ = 0.40; AIC = 27,756; Bayesian information criterion [BIC] = 27,873), and that on the non-paralyzed side was three (N = 6648; $R^2$ = 0.41; AIC = 15,673; BIC = 15,754). For the return motion, the number of clusters on the paralyzed side was four (N = 11,574; $R^2$ = 0.34; AIC = 30,581; BIC = 30,699), and that on the non-paralyzed side was four (N = 7429; $R^2$ = 0.50; AIC = 14,974; BIC = 15,085; Table 6).

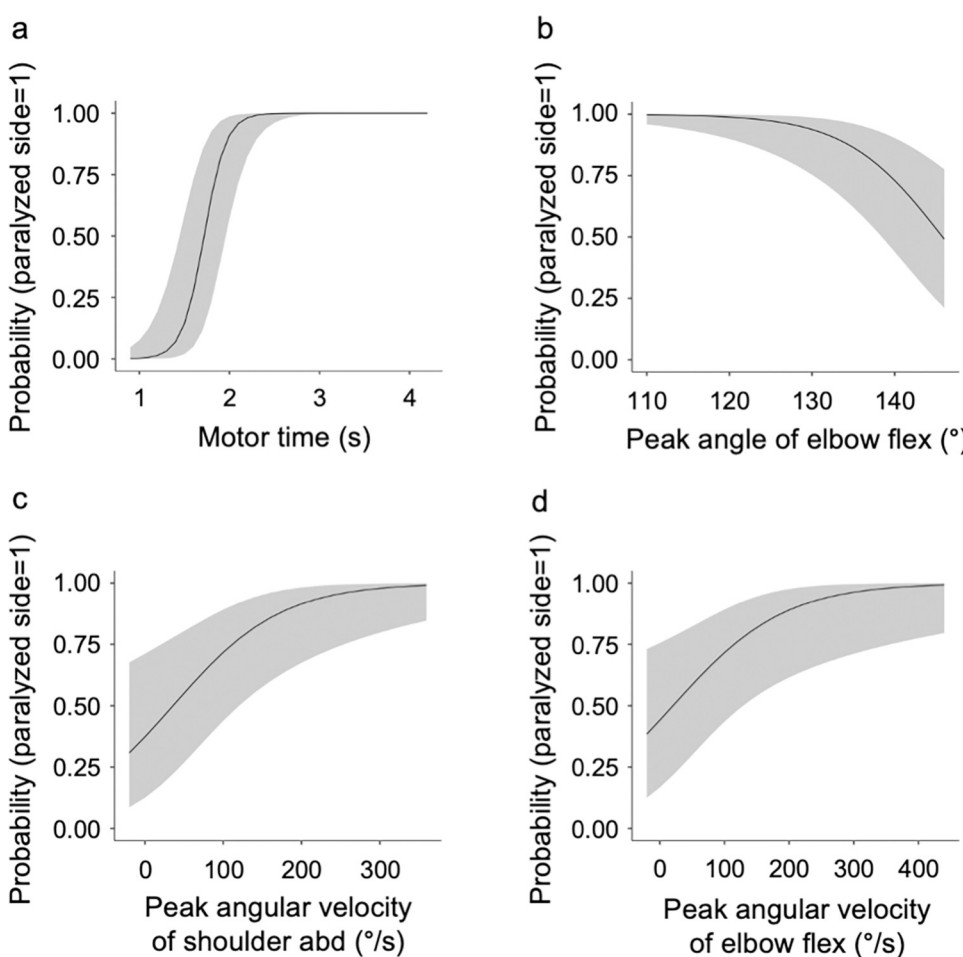

**Fig 8. Results of the binomial logistic regression analysis for the features of reaching motion from the occiput to the starting position.** (a) Motor time, (b) peak angle of elbow flex, (c) peak angular velocity of shoulder abd, and (d) peak angular velocity of elbow flex. abd, abduction; flex, flexion. The vertical axis of the figure is the probability, and the closer the value is to 1.00, the higher the probability that the upper limb is on the paralyzed side. If the motion time was 2.0 s on the return motion, the measured upper limb had approximately 80% probability of being on the paralyzed side.

**Table 5. Results of the receiver operating characteristic curve analysis.**

| Scale | | Cutoff point | Sensitivity (%) | Specificity (%) | Youden's index | AUC |
|---|---|---|---|---|---|---|
| Reaching the occiput | | | | | | |
| Motor time (s) | | 1.6 | 80 | 98 | 0.78 | 0.96 |
| Peak angle (°) | Shoulder flex | 55 | 84 | 44 | 0.28 | 0.63 |
| | Elbow flex | 145 | 96 | 4 | 0.00 | 0.28 |
| Reaching from the occiput to the starting limb position | | | | | | |
| Motor time (s) | | 1.6 | 78 | 90 | 0.68 | 0.92 |
| Peak angle (°) | Elbow flex | 145 | 100 | 6 | 0.06 | 0.29 |
| Peak angular velocity (°/s) | Shoulder abd | 53 | 84 | 48 | 0.32 | 0.65 |
| | Elbow flex | 62 | 74 | 56 | 0.30 | 0.67 |

abd, abduction; AUC, area under the curve; flex, flexion.

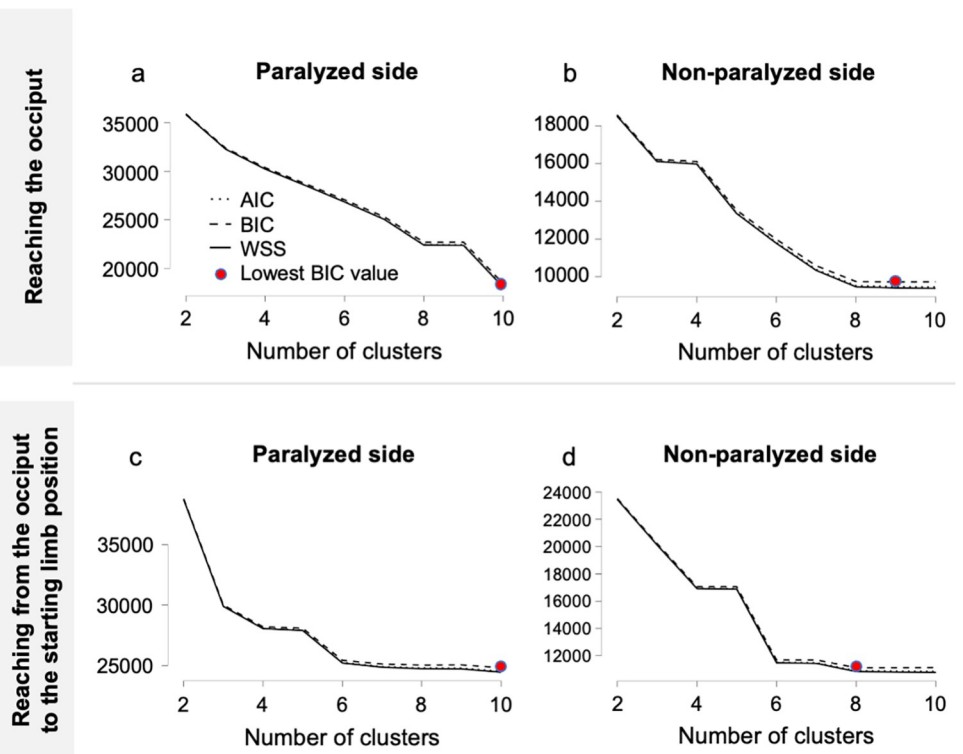

**Fig 9. Random forest clustering of the elbow method plot.** A plot of the intra-cluster sums of squares of errors for each cluster is shown in the figure. The number of clusters is (a) four for the paralyzed side and (b) three for the non-paralyzed side of the reaching motion to the occiput; and (c) four for the paralyzed side and (d) four for the non-paralyzed side of the reaching motion from the occiput to the starting position. AIC, Akaike's information criterion; BIC, Bayesian information criterion; WSS, within sum of squares.

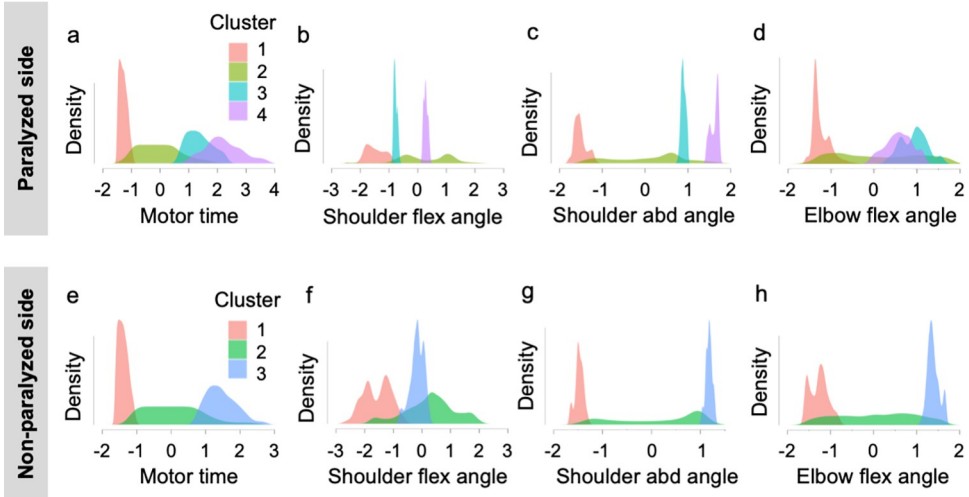

**Fig 10. Random forest clustering of the features of reaching motion to the occiput.** The upper panel shows the clustering structure of the (a) motor time, (b) shoulder flex angle, (c) shoulder abd angle, and (d) elbow flex angle on the paralyzed side. In the lower panel, the clustering structure of the (e) motor time, (f) shoulder flex angle, (g) shoulder abd angle, and (h) elbow flex angle for the non-paralyzed side is shown. The horizontal axis of the figure indicates the Z-values. The areas of each cluster composed of densities and absolute Z-values of each parameter are all equal. abd, abduction; flex, flexion.

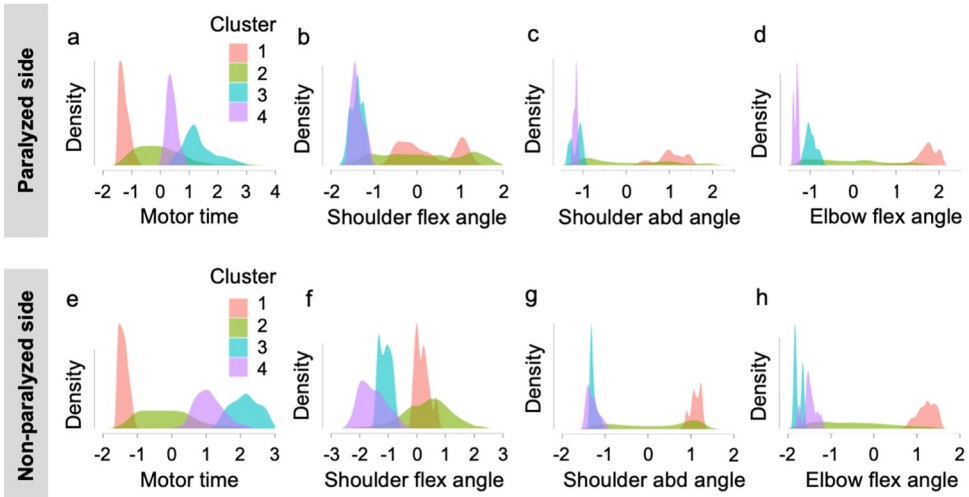

**Fig 11. Random forest clustering of the features of reaching motion from the occiput to the starting position.** The upper panel shows the clustering structure of the (a) motor time, (b) shoulder flex angle, (c) shoulder abd angle, and (d) elbow flex angle on the paralyzed side. In the lower panel, the clustering structure of the (e) motor time, (f) shoulder flex angle, (g) shoulder abd angle, and (h) elbow flex angle for the non-paralyzed side is shown. The horizontal axis of the figure indicates the Z-values. The areas of each cluster composed of densities and absolute Z-values of each parameter are all equal. abd, abduction; flex, flexion.

## Discussion

In this study, assuming that there is no left-right difference in the movement pattern between the dominant and non-dominant upper limbs for reaching the back of the head with the hand in normal individuals without motor paralysis, we hypothesized that movement time, joint angles, and angular velocity while performing the movement of reaching the back of the head with the hand in patients with chronic phase mild hemiplegia will differ between the paralyzed and non-paralyzed upper limbs. A repeated measures multivariate analysis of covariance was performed to test the hypothesis that motor time, joint angles, and angular velocity while doing movements to reach the occiput differ between paralyzed and non-paralyzed upper limbs. During the outward reaching motion, the paralyzed side showed a significantly longer motor time and significantly greater peak shoulder flexion angle. During the return reaching motion, the motor time was significantly longer on the paralyzed side, and the peak angular velocities of elbow flexion, shoulder flexion, and shoulder abduction were significantly greater on the paralyzed side. In contrast, the peak elbow flexion angle was significantly greater on the non-paralyzed side. Next, to detect the cutoff values of the motor features that discriminate

**Table 6. Results of random forest clustering.**

| Measuring side | Clusters | N | $R^2$ | AIC | BIC |
|---|---|---|---|---|---|
| Reaching motion to the occiput | | | | | |
| Paralyzed side | 4 | 11542 | 0.40 | 27756.22 | 27873.88 |
| Non-paralyzed side | 3 | 6648 | 0.41 | 15673.33 | 15754.96 |
| Reaching motion from the occiput to the starting position | | | | | |
| Paralyzed side | 4 | 11574 | 0.34 | 30581.45 | 30699.15 |
| Non-paralyzed side | 4 | 7429 | 0.50 | 14974.92 | 15085.53 |

AIC, Akaike's information criterion; BIC, Bayesian information criterion.

between the paralyzed and non-paralyzed sides, binomial logistic regression analysis was conducted on the features fitted to the models for the paralyzed and non-paralyzed sides using repeated measures multivariate analysis of covariance. For the features fitted to the regression model, the cutoff values discriminating between the paralyzed and non-paralyzed sides were calculated using the ROC curve. As a result, during the outward reaching motion, the motor time and peak angles of shoulder flexion and elbow flexion were calculated to be 1.6 s, 55˚, and 145˚, respectively. During the return reaching motion, the motor time, peak elbow flexion angle, and peak angular velocities of shoulder abduction and elbow flexion were calculated to be 1.6 s, 145˚, 53˚/s, and 62˚/s, respectively. The calculated cutoff values can be used as the target values for treatment to improve the reaching motion to the occiput in patients with mild hemiplegia. The quality of upper extremity movement can be improved by exercises based on kinematic values [52]. Recently, kinematic biofeedback has been developed as a new rehabilitation therapy [53]. This is a treatment in which patients are given exercises while alternative proprioceptive feedback is returned to them based on the results measured by 3DMA and electromyography. Although biofeedback has been introduced to restore the function of reaching movements in stroke patients, the effectiveness of the treatment has not been fully tested, and solid recommendations have not been provided [54]. The results of this study are useful as reference data for the application of biofeedback, robotics rehabilitation, and other technologies and are expected to contribute to the development of therapies for restoring the patients' functions and activity limitations.

Among the cutoff values detected, the kinematic feature with the highest Youden's index was the motor time for both the outward and return reaching motions. Previous studies have reported that patients with stroke have a longer motor time, and the most difficult subitem in the FMA-UE is the motor time measured in Part D [55, 56]. In Part D of the FMA-UE, the subject is asked to flex the elbow joint and touch the nose with the index finger five times from the position of shoulder abduction and elbow extension, which is a movement task that requires joint motion coordination which is a movement task that requires, just like reaching to the back of the head, joint motion coordination. Motor time is presumed to be an index that determines whether the reaching motion to the occiput in a patient is a near-normal motion. In this study, the cutoff value for the peak elbow flexion angle was 145˚ for both the outward and return reaching motions. Hairdressing with a reaching motion to the back of the head requires the highest elbow joint flexion angle among the verified daily activities [57]. Hence, we inferred that sufficient elbow joint flexion is important to reach the occiput. In the outward reaching motion, the peak angle was significantly larger on the paralyzed side during shoulder flexion and on the non-paralyzed side during elbow flexion. Reaching the paralyzed side in patients with stroke has been reported to result in decreased coordination of the shoulder and elbow joint movements [58]. In the outward motion of reaching training, the goal of the exercise is to shorten the motor time and increase the elbow flexion angle more than the shoulder flexion angle. The cutoff values for angular velocities of the joints were determined in the return reaching motion. When the hand reached the occiput, the upper limb was in abduction at the shoulder joint and flexion at the elbow joint, which is the joint motion pattern of the flexor muscles of the upper limb that appears after a stroke [59, 60]. A previous study reported that muscle spasticity affects joint movement coordination in reaching exercises using the paralyzed upper limb in patients with stroke [61]. The limb position, in which the hand reaches the occiput, induces a joint movement pattern of the upper limb and tends to increase the muscle tone of the flexor muscle group. Therefore, it may be difficult to control the angular velocity of the joint when the upper limb is lowered. In the return motion of reaching training, the practice goal is to shorten the movement time and decrease the angular velocity of shoulder abduction and elbow flexion.

In the present study, changes in the motion time and the joint angle of the paralyzed and non-paralyzed sides during the reaching motion to the occipital region were analyzed using random forest clustering. Using these results as a reference, one practical strategy is to approximate the cluster on the paralyzed side to the cluster on the non-paralyzed side of the patient. The motor time of cluster 1 on the paralyzed side in the reaching motion to the occiput was shorter than the cutoff value. The joint angles of cluster 1 in the paralyzed group were similar to those of cluster 1 in the non-paralyzed group. The motor time of cluster 2 on the paralyzed side was within and longer than the cutoff value. Cluster 2 on the paralyzed side exhibited a variety of joint angles similar to those of cluster 2 on the non-paralyzed side. For patients in cluster 2 on the paralyzed side whose motor time was approximately 1.6 s, we suggested that one of the exercises should be to shorten the motor time by reducing the variation in joint angles such that the hand can reach the occiput in the shortest distance. The motor times of patients in paralytic clusters 3 and 4 were longer than the cutoff values. Hence, we recommended that patients in paralytic clusters 3 and 4 practice joint movement patterns to approach the values of those in non-paralytic clusters 1 or 3. When targeting cluster 1 on the non-paralytic side, a method to reduce the changes in the joint angle and shorten the motor time exists. When targeting cluster 3 on the non-paralyzed side, patients in cluster 3 on the paralyzed side should be trained by increasing the angular changes in shoulder flexion and elbow flexion while aiming for a motor time near the cutoff value. For patients in cluster 4 on the paralyzed side, the motor time was aimed to be near the cutoff value, and the goals of practice were to decrease the angular change in shoulder abduction and increase the angular change in elbow flexion (S1 Table).

In the reaching motion from the occiput to the starting limb, the motion time of cluster 1 on the paralyzed side was shorter than the cutoff value, and, as in the reaching motion to the occiput, the motion pattern of cluster 1 on the paralyzed side was similar to that on the non-paralyzed side. The motor time of cluster 2 on the paralyzed side was within or near the cutoff value. For cluster 2 on the paralyzed side, one of the practice goals was to shorten the motor time. The motor times of clusters 3 and 4 on the paralyzed side were longer than the cutoff values. The joint angles of clusters 3 and 4 on the paralyzed side were similar to those of clusters 3 and 4 on the non-paralyzed side. These results suggest that there is no difference in the motion pattern between the paralyzed and non-paralyzed sides when reaching from the occiput to the starting position. For the reaching motion from the occiput to the starting position, we proposed a practice to control the angular velocity of shoulder abduction and elbow flexion, with the target motor time within or near the cutoff value (S1 Table).

This study had several limitations. First, the participants were patients with mild hemiplegia, whose condition persisted for more than 6 months; therefore, the obtained results should be used with caution in patients with early onset of the disease. Furthermore, eight of 10 participants in this study had a mAs score of 1 point, and we did not include patients with significant spasticity. Patients with severe motor paralysis are more likely to have strong muscle spasticity and exhibit joint movement patterns in the upper limbs [62]. Patients with severe motor paralysis who are unable to reach the occiput with their hands may have difficulties in practicing joint movement patterns based on the results of this study. Second, we did not compare the movements of the paralyzed upper limbs of patients with stroke with those of healthy participants. Patients with post-stroke hemiplegia also suffer from motor paralysis of the trunk and lower limb muscles, their balance function may be impaired [63, 64], and, during the reaching motion, the trunk and lower limb functions influence the upper limb joint motion [65, 66]. Therefore, in this study, the sitting posture at the time of measurement was defined in detail, and the displacement of the markers attached to the trunk was used as an adjustment variable in the statistical analysis. Based on the results of the FMA-LE, BBS, and FRT, the participants

in this study had mild motor paralysis of the lower limbs and good trunk and balance function. However, how the lower limb and trunk movements as well as the balance strategies for sitting and holding positions were involved in upper limb movements requires further analysis. To target the upper limb movements of individuals with no motor paralysis, an analysis of the upper limb movements of healthy participants is required. Third, in the present analysis, it was not proven that the within-subject variability (left-right difference) was smaller than the between-subject variability. Because normal subjects have clearly asymmetric motor strategies in many movements, the criterion for a paralyzed movement is not necessarily the corresponding movement of the non-paralyzed limb. To resolve this limitation, the present upper limb task should be analyzed in the movements of healthy subjects to verify that it is symmetrical [67]. Fourth, the distal joints of the upper extremities were not analyzed. During the reaching motion, limitations in the joint range of motion in the forearm and wrist joints affect the shoulder joint motion [68]. In this study, the joint motions of the shoulder and elbow joints were analyzed; however, coordination with the distal joints of the upper limb was not verified. Fifth, the motor tasks were measured at patients' comfortable speeds. When the movement is measured at a specified speed, it may cause changes in the joint motion and compensatory movement strategies are needed [69, 70]. Sixth, the cutoff values for motion time and joint angle were calculated based on the results measured using 3DMA. However, these cutoff values should be used as reference data because measurement and analysis using 3DMA are complicated and unlikely to be performed in clinical practice. It remains unclear whether the practice of movement patterns performed in accordance with the optimal movement time and angular velocity on the non-paralyzed side can restore motor paralysis. Further validation is needed to consider these limitations to clinically utilize the practice methods and target values to reproduce movements equivalent to those of the non-paralyzed side without the appearance of spasticity or compensatory movements when upper limb motor paralysis is mild.

## Conclusions

In this study, the kinematic features and cutoff values while reaching the back of the head in patients with mild hemiplegia in the chronic phase were detected, and patterns of changes in the joint angle were analyzed. Based on our findings, when patients with hemiplegia who can reach the back of the head practice with the goal of smoother upper limb motion, the motion patterns of the non-paralyzed upper limb can be referenced to set the target values of motion time, joint angle, and angular velocity. The motor time of the paralyzed upper limb was measured, and the corresponding clusters were referred to from the results of random forest clustering. The results of this study will be used to plan motor practice strategies that trace the pattern of joint angles of the upper limb without paralysis. Moreover, the results obtained in this study can be used as a reference to devise effective practice methods to improve the reaching motion to the back of the head.

## Supporting information

**S1 Table. Training methods devised from the pattern analysis results.**
(DOCX)

## Acknowledgments

We would like to thank the occupational therapists at the Department of Rehabilitation Medicine, Jikei University Hospital, for their cooperation in obtaining the data for this study. Moreover, we would like to thank Prof. Michito Namekawa and Prof. Satoshi Kido, Department of

Rehabilitation, Graduate School of Health Science, Saitama Prefectural University, for their clinical advice for the successful completion of this study.

## Author Contributions

**Conceptualization:** Daigo Sakamoto, Toyohiro Hamaguchi, Masahiro Abo.

**Data curation:** Daigo Sakamoto, Toyohiro Hamaguchi, Keisuke Kubota, Ryota Suwabe.

**Formal analysis:** Daigo Sakamoto, Toyohiro Hamaguchi, Keisuke Kubota, Ryota Suwabe.

**Investigation:** Daigo Sakamoto.

**Methodology:** Daigo Sakamoto, Toyohiro Hamaguchi, Naohiko Kanemura, Takashi Yasojima, Masahiro Abo.

**Project administration:** Yasuhide Nakayama, Masahiro Abo.

**Resources:** Yasuhide Nakayama, Masahiro Abo.

**Software:** Keisuke Kubota, Ryota Suwabe, Masahiro Abo.

**Supervision:** Toyohiro Hamaguchi, Naohiko Kanemura, Takashi Yasojima, Yasuhide Nakayama, Masahiro Abo.

**Validation:** Naohiko Kanemura, Takashi Yasojima, Yasuhide Nakayama.

**Visualization:** Daigo Sakamoto, Toyohiro Hamaguchi.

**Writing – original draft:** Daigo Sakamoto.

**Writing – review & editing:** Toyohiro Hamaguchi, Naohiko Kanemura, Takashi Yasojima, Keisuke Kubota, Ryota Suwabe, Yasuhide Nakayama, Masahiro Abo.

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
