## [Decision Letter · Decision Letter 0]

1 Feb 2024

PONE-D-23-37600Feature analysis of joint motion while reaching the occiput in patients with mild hemiplegia: a cross-sectional studyPLOS ONE

Dear Dr. Hamaguchi,

Thank you for submitting your manuscript to PLOS ONE. After careful consideration, we feel that it has merit but does not fully meet PLOS ONE’s publication criteria as it currently stands. Therefore, we invite you to submit a revised version of the manuscript that addresses the points raised during the review process.

**ACADEMIC EDITOR:**Although Authors recognized merits in the study, some major and minor points have to be carefully addressed before acceptance for publication.

We look forward to receiving your revised manuscript.

Kind regards,

Andrea Tigrini, Ph.D.

Academic Editor

PLOS ONE

Additional Editor Comments:

Although Authors recognized merits in the study, some major and minor points have to be carefully addressed before acceptance for publication.

Reviewers' comments:

Reviewer's Responses to Questions

**Comments to the Author**

1. Is the manuscript technically sound, and do the data support the conclusions?

Reviewer #1: Partly

Reviewer #2: Yes

Reviewer #3: Partly

2. Has the statistical analysis been performed appropriately and rigorously? 

Reviewer #1: I Don't Know

Reviewer #2: Yes

Reviewer #3: I Don't Know

3. Have the authors made all data underlying the findings in their manuscript fully available?

Reviewer #1: Yes

Reviewer #2: No

Reviewer #3: Yes

4. Is the manuscript presented in an intelligible fashion and written in standard English?

Reviewer #1: Yes

Reviewer #2: Yes

Reviewer #3: No

5. Review Comments to the Author

Reviewer #1: The article “Feature analysis of joint motion while reaching the occiput in patients with mild hemiplegia: a cross-sectional study” aims at better characterizing motor strategy in post-stroke patients during one peculiar task, i.e. reaching the back of the head. The protocol is well described and the data analysis method proposed is interesting to go deeper into the understanding.

My main concern is that the goal is too fuzzy as the rationale is too light. It is mainly saying: there are too few articles providing raw data regarding that particular task (reaching the back of the head with one hand) so we will perform 3D measures and provide an analysis of raw data. Why this task rather than another? Several tasks implying to bring the hand in the posterior side of the body might be as important as this one, e.g. mid-back or low-back. What is useful for to better know the specific characteristics of post-stroke people motor strategy to achieve this particular task? I am not saying that is useless, only that you do not explain clearly why it is useful in a clinical perspective.

Minor points

L50: “to improve motor paralysis” sounds a bit strange. It should rather be “to improve motor capacities”

L52: “Reaching is a fundamental movement of the upper limb, with the goal of reaching the target point” please rephrase. It is a loop; you define “reaching” by “reaching”

L81: “because of the discrepancy between the results and the upper limb movements in daily activities” I do not understand what could be a discrepancy between results (which ones?) and movements. Do you mean movements performed in the protocol are too far from ADL? Or do you mean the results of several studies are not consistent? The link with the next sentence, “As the tasks are measured in a manner that is close to daily life situations…”, is also not clear. You’ve just said that they are too far from properly represent ADL and then you write they are close. Is it far or close?

L85: “the kinematic characteristics of patients with stroke with mild motor paralysis have not been verified” What do you mean by “verified”? We verify hypotheses but how can we verify characteristics?

L90: “When upper limb motor tasks are measured using a three-dimensional motion analyzer, the motions of the paralyzed and non-paralyzed sides should be compared” I do not agree with this. The reference for one given motion of paretic limb is not always the corresponding motion of non-paretic limb. Healthy people have obviously asymmetric motor strategies for some many tasks. Comparing dominant and non-dominant limbs’ motions is not the quantification of the effect of a deficiency. Your sentence is only true if you have previously demonstrated that, for one given task, motor strategy is symmetric in healthy population, meaning intra-subject variability (left w.r.t right) is lower than inter-subjects’ one.

L92: “however, the non-paralytic upper limb, which does not show obvious functional impairment, can be used in clinical situations for comparison with the paralytic upper limb” I do not fully agree with this sentence either. Compensation strategies in non-paretic limb are neglected.

L111: “while reaching the hand to the back of the head” is not clear. Patient are not reaching the hand but reaching the back of their head with their hand

P128: Rather than exclusion criteria, I would recommend to use non-inclusion criteria. To me exclusion criteria stand for situations occurring after inclusion that force experimenters to withdraw one participant from the study while he/she has already started the experiment. On the opposite, non-inclusion means that this person has never been a participant, i.e. has never started the experiment.

L152 and following: what is the need for markers on lower limbs if participants remain seated and if you are only analyzing upper body joints motions?

L161: what do you mean by “thermal cameras”? Is it infrared cameras?

L161: what is the goal of the “landmark placed 5m away from the participants”? On L188 you explain that participants had to maintain gaze on this landmark

L179: “elbow in full extension, forearm in mid-extension” This is a bit strange. Once you are speaking about a joint then about a segment. By forearm, did you actually mean wrist? Why was it in mid-extension rather than in neutral posture? Furthermore, was this posture (elbow extension, wrist mid-extension, …) always reachable for patients, especially with paretic limb? Considering the usual triple flexion posture observed in that limb, it seems that the starting position you ask is hardly reachable. Your following sentence, i.e. relax the muscles as much as possible, is indeed a more valuable instruction. Fig 2b does not illustrate at all this instruction. L195, you precise “the examiner verified whether the participants were in the correct starting position”. This is a bit contradictory with sentence on L181-183. What was the actual tolerance for the starting position?

L279: “to test the hypothesis that the kinematic characteristics […] differ between the paralyzed and non-paralyzed upper limbs”. First, this hypothesis must be better explained in introduction section. Then, this hypothesis is much too large. Even if you specify just after the independent variables taken into account, such a large hypothesis allows that at least one parameter will differ. Finally, and once again, a difference between paretic and non-paretic limbs in this population only means something under the hypothesis that the corresponding difference (dominant vs. non-dominant) does not exist in healthy population.

L310: what is IG in equation 2? Moreover, this equation could be re-written considering paretic/non-paretic limbs rather than left/right.

L332: “Of the 88 patients, 75 were excluded” How this is possible? What is the reason for such a discrepancy between first screening and inclusion? Were the non-included patients particularly affected or was the FMA or ARAT threshold too high?

L341: “All the participants had right hemiplegia” Was it a coincidence or is there something in the inclusion/non-inclusion criteria that favor inclusion of right hemiplegia? It is a bit surprising as they were also all right-dominant before stroke and therefore potentially more affected by hemiplegia.

Table 2: variability in peak angular velocities is very high. Did you check normality for these parameters before comparing sides?

Figures are hardly readable

Reviewer #2: The manuscript is well written and I identify a major conercern that deserves to be strenghten. In my opinion is very important to stress the rationale bheind the study and the application that a study with a strong kinesiological background may have in relation to the development of modern assistive technologies. For this reason I suggest the Authros to betterframe this aspect both in the introduction and in the discussion. Under this perspective I suggest Authros to rviewe the following recent manuscripts:

-"On the Decoding of Shoulder Joint Intent of Motion from Transient EMG: Feature Evaluation and Classification." IEEE Transactions on Medical Robotics and Bionics (2023)".

-"Rehabilitation with Kinematic Biofeedback Improves Shoulder Function in Patients Surgically Treated for Rotator Cuff Tear: Indications from a Randomized Controlled Trial." (2023).

Reviewer #3: General comments

This manuscript suggests kinematic values for reaching the occiput, aiming to enhance patients' daily movements. This approach holds promise for upper-limb kinematic analysis in clinical settings. However, several concerns need addressing before publication. Primarily, redundancies in the descriptions have been identified. It is advisable to review the use of abbreviations and incorporate subheadings within the Materials and Methods section. Additional comments are outlined below:

Abstract

L 26, P 2: The background of this study should be explained initially.

L 34, P 2: In terms of methods, it could be added that the random forest clustering analysis was conducted.

Introduction

L 45, P 2: In many cases, kinematic analysis involves comparing conditions. However, this study aimed to establish kinematic cutoff values and motion patterns for the upper limb using random forest clustering. The discussion includes the application of these findings in clinical practice. The Introduction would outline the novelty of this approach and explores its clinical implications.

L 75, P 4: Please replace "Three-dimensional motion analysis" with "3DMA" for improved readability.

L 76 – 77, P 4: Describing the system seems unnecessary.

L 82, P 4: What does "the tasks" refer to?

L 89, P 4: If possible, consider dividing this paragraph into two sections to enhance readability.

L 94, P 4: Delete the sentence "In a recent study…" to avoid redundancy with the following sentence, as the inertial sensors used were 3DMA.

L 102, P 5: Comparing paralyzed and non-paralyzed upper limbs is crucial for the clinical perspective of this study, which is its primary aim.

Materials and methods

L 120, P 5: “Patients with post-stroke hemiplegia who attended the Department of…” Are they outpatients?

L 130 – 131, P 6: "Higher brain dysfunction," this term is developed and commonly used only in Japan. For instance, you may rephrase it to "cognitive impairment after a stroke."

L 145, P 6: “the sample size,” is this Hedge’s d?

L 154, P 7: Please provide the reference for "the Plug-in Gait marker model" here.

L 154 - 157, P 7: "The examiner made the participants sit… …and their upper limbs drooped." I did not understand what postures were set for. Is there marker setting or task measurement?

L 161, P 7: What is "a landmark" used for?

L 164, P 7: Captions for figures must include a detailed description. For example, the position of the circles indicates the infrared markers attached to the participants’ bodies to construct the Plug-in Gait model.

L 175, P 8: Does the arrow and circle in the illustration indicate that the participants maintained their viewpoints on the landmark while performing the task?

L 157 – 162, P 7: This section explains the setting of the starting measurements. The descriptions of the marker setting above and below this section are divided.

L 179, P 8: “forearm in mid-extension.” Is this correct?

L 217, P 9 – L 224, P 10: Were the peak values picked up first? Was the data for the angles, angular velocities, and displacements average of five times?

L 224–225, P 10: Were the peak values of the joint angles and angular velocities?

L 281 – 282, P 12: “The dependent variable” and “the independent variables.” Please check whether these words must be exchanged.

L 288, P 12: Please explain why the cutoff values were calculated.

Results

L 327 and 338, P 14: Consider merging subheadings "Participants" and "Descriptive data" into a single subheading titled "Participants."

L 342 and 343, P 14: Review the necessity of including the phrase "of FMA-UE."

L 355, P 17 and L 368, P 20: Merge subheadings "Outcome data" and "Detection of motion features" into a unified subheading, such as "Detection of paralyzed upper limb with motion data."

L 359, P 18 and L 360, P 19: In Tables 2 and 3, examine the effect size of the Para and Non-para sides comparison using MANCOVA. The sample size (Hedges’ g = 0.4) was chosen to detect large effects.

L 367, P 20: Consider omitting the subheading "Main findings" to enhance readability.

L 394 and 398, P 24: Clarify whether the probability of a non-paralyzed limb is 0. Provide an explanation on interpreting Figs 6 and 7.

L 422 – 426, P 26: If feasible, elaborate on the features of the clusters focused on. Readers may find it challenging to interpret these graphs alone (Figs 9 and 10).

Discussion

L 467 – 470, P 28: The results for the peak angle of elbow flexion are missing. Please verify them.

L 475 – 477, P 28: The result of the returning motion is also incorrect.

L 480, P 29: Your calculated cutoff values and motion features may serve as the target values for treating reaching occiput. Treating with these kinematic values might enhance the quality of upper limb movements (Prange‑Lasonder et al., J NeuroEngineering Rehabil, 2021, 18:162). Therefore, what improvements in movement changes can be observed in patients? Express these clinical implications and emphasize the need for future studies, as discussed in the Introduction.

L 485, P 29: What does "Part D" refer to?

L 496, P 29: In this study, you calculated angular velocity, not angular acceleration.

L 541, P 31: The cut-off values proposed by this study were calculated at 0.1 second and 1 degree. These values were measured using 3DMA. However, this method is not commonly employed clinically. This point should be addressed in the Limitations section.

6. PLOS authors have the option to publish the peer review history of their article (what does this mean?). If published, this will include your full peer review and any attached files.

Reviewer #1: No

Reviewer #2: No

Reviewer #3: **Yes: **Jun Nakatake

---

## [Author Response · Author response to Decision Letter 0]

13 Mar 2024

Reviewer #1

The article “Feature analysis of joint motion while reaching the occiput in patients with mild hemiplegia: a cross-sectional study” aims at better characterizing motor strategy in post-stroke patients during one peculiar task, i.e. reaching the back of the head. The protocol is well described and the data analysis method proposed is interesting to go deeper into the understanding.

Comment 1

My main concern is that the goal is too fuzzy as the rationale is too light. It is mainly saying: there are too few articles providing raw data regarding that particular task (reaching the back of the head with one hand) so we will perform 3D measures and provide an analysis of raw data. Why this task rather than another? Several tasks implying to bring the hand in the posterior side of the body might be as important as this one, e.g. mid-back or low-back. What is useful for to better know the specific characteristics of post-stroke people motor strategy to achieve this particular task? I am not saying that is useless, only that you do not explain clearly why it is useful in a clinical perspective. 

Response to comment 1

Thank you for your important comments. We have added the following explanation regarding the significance of this study.

“Reaching the back of the head requires a wide range of motion and coordination between the shoulder and elbow joints, which is a challenging task for patients with motor paralysis. Besides the occiput, the mouth, chest, abdomen, and lower back are other reaching target points for eating, dressing, and bathing. While important for self-care performance, reaching these target points is less difficult than reaching to occipital area, thus, patients with mild hemiplegia are often able to perform these reaches adequately [7,8]. The ability to smoothly reach the back of the head, which is difficult for hemiplegics, is beneficial to their ADLs.” (Introduction, lines 62–69)

“Due to a ceiling effect, upper extremity function scores used in clinical practice do not measure the recovery in patients with mild motor paralysis [14,15]. When planning a change in joint movement pattern exercises, or to shorten their duration, attempting to use the clinically obtained scores is challenging. To the best of our knowledge, a kinematic index that quantitatively expresses the "reach to the back of the head" does not exist. Moreover, a clear target value to determine if the reach has been sufficiently recovered in patients with mild motor paralysis is lacking.” (Introduction, lines 81–87)

Comment 2

L50: “to improve motor paralysis” sounds a bit strange. It should rather be “to improve motor capacities” 

Response to comment 2

We have revised the text as follows. 

“As motor paralysis of the upper limbs and fingers limits patients' activities of daily living (ADLs) and decreases their quality of life (QOL), patients are provided with continuous rehabilitation to improve their motor capacities [3,4].” (Introduction, lines 52–54)

Comment 3

L52: “Reaching is a fundamental movement of the upper limb, with the goal of reaching the target point” please rephrase. It is a loop; you define “reaching” by “reaching” 

Response to comment 3

We have made modifications to the definition of Reaching.

“Reaching is a basic movement of the upper extremity aimed at extending the arm toward a designated target in order to touch or grasp something [5]. ” (Introduction, lines 55–57)

Comment 4

L81: “because of the discrepancy between the results and the upper limb movements in daily activities” I do not understand what could be a discrepancy between results (which ones?) and movements. Do you mean movements performed in the protocol are too far from ADL? Or do you mean the results of several studies are not consistent? The link with the next sentence, “As the tasks are measured in a manner that is close to daily life situations…”, is also not clear. You’ve just said that they are too far from properly represent ADL and then you write they are close. Is it far or close? 

Response to comment 4

We apologize for the misunderstanding. We did not want to emphasize the discrepancies between studies. We wished to highlight the usefulness of measuring movements that are similar to ADLs. We have revised the text in this regard.

“Kinematic characteristics calculated by analyzing movements related to daily life are strongly related to the patient's activity capacity, and are valid indicators of patient treatment in clinical practice [21–24].” (Introduction, lines 93-96)

Comment 5

L85: “the kinematic characteristics of patients with stroke with mild motor paralysis have not been verified” What do you mean by “verified”? We verify hypotheses but how can we verify characteristics? 

Response to comment 5

We have not explained this point well enough. We have added the following text.

“Several studies have evaluated healthy participants and patients with orthopedic diseases in terms of their ability to reach the occipital region; however, kinematic characteristics such as motor time, angular change of joints, and angular velocity of joints have not been validated in stroke patients with mild motor paralysis [9,10,25]. This issue can be analyzed using 3DMA to provide quantified data.” (Introduction section, lines 96–100)

Comment 6

L90: “When upper limb motor tasks are measured using a three-dimensional motion analyzer, the motions of the paralyzed and non-paralyzed sides should be compared” I do not agree with this. The reference for one given motion of paretic limb is not always the corresponding motion of non-paretic limb. Healthy people have obviously asymmetric motor strategies for some many tasks. Comparing dominant and non-dominant limbs’ motions is not the quantification of the effect of a deficiency. Your sentence is only true if you have previously demonstrated that, for one given task, motor strategy is symmetric in healthy population, meaning intra-subject variability (left w.r.t right) is lower than inter-subjects’ one. 

Response to comment 6

Thank you for your helpful remarks on this important aspect of our study. We should have been careful in our discussion about comparing the paralyzed side with the non-paralyzed side. We have revised the manuscript and added the following text in response to your comments.

“Upper extremity motor tasks can be analyzed using 3DMA by comparing healthy subjects and patients; however, in clinical situations, movements on the paralyzed and nonparalyzed side of the patient may be compared [26]. The movements of the nonparalytic upper limb in patients with hemiplegia after a stroke differ from those of normal participants, and the reference for a paralyzed movement is not necessarily the corresponding movement of the nonparalyzed limb. Although the nonparalyzed upper limb may also have compensatory strategies for movement, a comparison of the kinematic characteristics of the paralyzed and nonparalyzed upper limbs of patients with hemiplegia with mild impairment of limb and trunk function has been conducted to validate the calculation of the evaluation values that will be used as a reference for treatment [27,28].” (Introduction, lines 103–112)

“Third, in the present analysis, it was not proven that the within-subject variability (left-right difference) was smaller than the between-subject variability. Because normal subjects have clearly asymmetric motor strategies in many movements, the criterion for a paralyzed movement is not necessarily the corresponding movement of the nonparalyzed limb. To resolve this limitation, the present upper limb task should be analyzed in the movements of healthy subjects to verify that it is symmetrical [66].” (Discussion, lines 620–626)

Comment 7

L92: “however, the non-paralytic upper limb, which does not show obvious functional impairment, can be used in clinical situations for comparison with the paralytic upper limb” I do not fully agree with this sentence either. Compensation strategies in non-paretic limb are neglected. 

Response to comment 7

Thank you for pointing this out. The limitations of comparing the paralyzed and non-paralyzed sides in this study should have been clearly stated. We have added the following text along with the response to your Comment 6.

“Third, in the present analysis, it was not proven that the within-subject variability (left-right difference) was smaller than the between-subject variability. Because normal subjects have clearly asymmetric motor strategies in many movements, the criterion for a paralyzed movement is not necessarily the corresponding movement of the nonparalyzed limb. To resolve this limitation, the present upper limb task should be analyzed in the movements of healthy subjects to verify that it is symmetrical [66].” (Discussion, lines 620–626)

Comment 8

L111: “while reaching the hand to the back of the head” is not clear. Patient are not reaching the hand but reaching the back of their head with their hand. 

Response to comment 8

We have refered to your comments and modified the text as follows. 

“In this cross-sectional study, kinematic data of patients with post-stroke hemiplegia were obtained while reaching the back of their head with their hand, and the patient’s paralyzed and non-paralyzed upper limbs were compared.” (Materials and methods, Study design, lines 126–128)

Comment 9

P128: Rather than exclusion criteria, I would recommend to use non-inclusion criteria. To me exclusion criteria stand for situations occurring after inclusion that force experimenters to withdraw one participant from the study while he/she has already started the experiment. On the opposite, non-inclusion means that this person has never been a participant, i.e. has never started the experiment. 

Response to comment 9

Thank you for your important comments. We have decided to use the phrase "non-inclusion criteria" in the text revised the text accordingly. 

“The non-inclusion criteria were as follows: cases with a paralyzed hand not reaching the external occipital ridge with automatic movements; presence of a central nervous system disease other than stroke, orthopedic disease, mental disorder, cognitive impairment after stroke, dementia, visual field disorder, and ataxia at diagnosis; subluxation of the shoulder joint; pain in the joints of the upper limb or fingers during exercise; presence of a significant limitation in the joint range of motion in the upper limb; and completion of the occupational therapy intervention. Patients who met the study eligibility criteria but did not meet the non-inclusion criteria were asked to participate in the study, and those who provided consent were considered participants.” (Materials and methods, Participants, lines 143–151)

Comment 10

L152 and following: what is the need for markers on lower limbs if participants remain seated and if you are only analyzing upper body joints motions? 

Response to comment 10

As you pointed out, there is a model in which markers are placed only on the upper limbs. However, owing to the version of our analysis equipment, it was difficult to perform the analysis using such a model. Therefore, the Plug-in Gait model, one of the commonly used models, was selected for this study. The choice of this model did not impose significantly more time constraints on the participants than the upper extremity model, because the examiners were skilled at applying the markers.

Comment 11

L161: what do you mean by “thermal cameras”? Is it infrared cameras? 

Response to comment 11

Yes, we used infrared cameras in this study. We have modified the text as follows. 

“Six infrared cameras were placed on the ceiling of the room, and a landmark was placed 5 m away from the participants (Fig 2).” (Materials and methods, Experimental procedure, lines 178–179)

Comment 12

L161: what is the goal of the “landmark placed 5m away from the participants”? On L188 you explain that participants had to maintain gaze on this landmark. 

Response to comment 12

As you mention, we did not describe the purpose of the landmark. We have added a sentence in this regard.

“The landmark was placed to minimize neck and trunk movement while the participants performed the reaching task looking at it.” (Materials and methods, Experimental procedure, lines 179–180)

Comment 13

L179: “elbow in full extension, forearm in mid-extension” This is a bit strange. Once you are speaking about a joint then about a segment. By forearm, did you actually mean wrist? Why was it in mid-extension rather than in neutral posture? Furthermore, was this posture (elbow extension, wrist mid-extension, …) always reachable for patients, especially with paretic limb? Considering the usual triple flexion posture observed in that limb, it seems that the starting position you ask is hardly reachable. Your following sentence, i.e. relax the muscles as much as possible, is indeed a more valuable instruction. Fig 2b does not illustrate at all this instruction. L195, you precise “the examiner verified whether the participants were in the correct starting position”. This is a bit contradictory with sentence on L181-183. What was the actual tolerance for the starting position? L179: "elbow in full extension, forearm in mid-extension" 

Response to comment 13

Thank you very much for your important remarks. There was an error in the English wording in this sentence. The forearm was not mid-extension, but mid-position between supination and pronation. We have corrected the sentence as follows. 

 “The starting position of the upper limb on the measurement side was set with the elbow in full extension, fingers in extension, and the forearm in mid-position between supination and pronation; the forearm and fingers were not in contact with the chair.” (Materials and methods, Experimental procedure, lines 168–171)

The patients with mild motor paralysis who participated in this study were able to achieve the starting limb position easily in the first half of the measurement because only two patients had moderate spasticity, and none of them had significant joint contractures. Eight patients had a mAs of 1 and 2 had a mAs of 1+ in the biceps muscle. However, as you pointed out, some of them could not fully reach the starting position because of an increased muscle tone caused by the repeated measurements. In this respect, the participants were allowed to relax their muscles during the one-minute rest period to reproduce the starting limb position as much as possible. After the break was over, we allowed some participants to be unable to reach the starting position, for example, with a slight flexion of the elbow joint. We modified the text in this regard as follows. 

“Before each measurement, the examiner instructed the participants to attempt to achieve the starting position with the upper limb as much as possible. If a participant was unable to assume the starting position due to heightened muscle tone in their upper limb, causing a slight bending of the elbow joint, the procedure was still considered appropriate.” (Materials and methods, Reaching task, lines 215–218)

We added a Figure demonstrating the starting limb position during the measurement of the task, and the Figure that showed the measurement environment was cited in the text accordingly.

“The upper limb on the non-tested side was also placed in the same position (Fig 1).” (Materials and methods, Experimental procedure, lines 171–172) 

“Fig 1. Starting position during the measurement. (a) side view of the starting position and (b) starting position of the upper limb.” (Materials and methods, Reaching task, lines 182–183)

“Six infrared cameras were placed on the ceiling of the room, and a landmark was placed 5 m away from the participants (Fig 2).” (Materials and methods, Reaching task, lines 178–179)

“Fig 2. Measurement environment. (a) side view and (b) top view.” (Materials and methods, Reaching task, line 185) 

Comment 14 

L279: “to test the hypothesis that the kinematic characteristics […] differ between the paralyzed and non-paralyzed upper limbs”. First, this hypothesis must be better explained in introduction section. Then, this hypothesis is much too large. Even if y

---

## [Decision Letter · Decision Letter 1]

8 Apr 2024

PONE-D-23-37600R1Feature analysis of joint motion while reaching the occiput in patients with mild hemiplegia: a cross-sectional studyPLOS ONE

Dear Dr. Hamaguchi,

Thank you for submitting your manuscript to PLOS ONE. After careful consideration, we feel that it has merit but does not fully meet PLOS ONE’s publication criteria as it currently stands. Therefore, we invite you to submit a revised version of the manuscript that addresses the points raised during the review process.

**Authors addressed the majority of the concerns rised by the experts. However, some major points deserve to be still solved. Please carefully provide a detailed revision following the comments provided by the experts.**

We look forward to receiving your revised manuscript.

Kind regards,

Andrea Tigrini, Ph.D.

Academic Editor

PLOS ONE

Journal Requirements:

Additional Editor Comments:

Authors addressed the majority of the concerns rised by the experts. However, some major points deserve to be still solved. Please carefully provide a detailed revision following the comments provided by the experts.

Reviewers' comments:

Reviewer's Responses to Questions

**Comments to the Author**

1. If the authors have adequately addressed your comments raised in a previous round of review and you feel that this manuscript is now acceptable for publication, you may indicate that here to bypass the “Comments to the Author” section, enter your conflict of interest statement in the “Confidential to Editor” section, and submit your "Accept" recommendation.

Reviewer #1: All comments have been addressed

Reviewer #2: All comments have been addressed

Reviewer #3: (No Response)

2. Is the manuscript technically sound, and do the data support the conclusions?

Reviewer #1: (No Response)

Reviewer #2: Yes

Reviewer #3: Partly

3. Has the statistical analysis been performed appropriately and rigorously? 

Reviewer #1: (No Response)

Reviewer #2: Yes

Reviewer #3: I Don't Know

4. Have the authors made all data underlying the findings in their manuscript fully available?

Reviewer #1: (No Response)

Reviewer #2: Yes

Reviewer #3: Yes

5. Is the manuscript presented in an intelligible fashion and written in standard English?

Reviewer #1: (No Response)

Reviewer #2: Yes

Reviewer #3: Yes

6. Review Comments to the Author

**Reviewer #1:** (No Response)

**Reviewer #2:** Authors addressed all my concerns, the manuscript has been consistently updated and I think that the paper can be published.

**Reviewer #3: **I am glad to have the opportunity to review the revised manuscript, which is significantly better than the previous version. However, my main concerns with the revised manuscript are data processing and the statistical tests employed. The authors should address those before publication.

L4 and L117–119: The comparison of paralyzed and non-paralyzed upper limbs in this study design is a limitation despite being a clinically focused point. Mentioning this in the title and the study objective will help your readership to focus on the target audience.

L158–161, P7: In G*power, I find this only in the exact test, and the effect size “g” is displayed there. What test family was used in your calculation? Also, what was your required sample size?

L185, P8: In this figure caption, please explain the landmark.

L223, P9: For illustrations in each trial, left is outward, and right is return motions. This should be added to the figure caption.

L246–251, P10–11: Did you input 50 data points into descriptive statistics and statistical tests? In general, each participant’s data is averaged first, then descriptive statistics are calculated, after which statistical tests are employed. In general, averaging trial data for each participant is performed to correct measurement errors. This is followed by statistical tests, which are performed to estimate the population.

L301, P13: Provide the effect size calculation and eligibility criteria.

L307, P13: To my knowledge, MANCOVA is used to compare independent samples, which means it is not applicable here.

7. PLOS authors have the option to publish the peer review history of their article (what does this mean?). If published, this will include your full peer review and any attached files.

Reviewer #1: No

Reviewer #2: No

Reviewer #3: No

---

## [Author Response · Author response to Decision Letter 1]

1 May 2024

Reviewer #3:

Comment 1

I am glad to have the opportunity to review the revised manuscript, which is significantly better than the previous version. However, my main concerns with the revised manuscript are data processing and the statistical tests employed. The authors should address those before publication.

Response to comment 1

We appreciate your insightful comments on the data processing and statistical testing methods in this study. We have referenced your comments and made revisions and additions to the manuscript accordingly.

Comment 2

L4 and L117–119: The comparison of paralyzed and non-paralyzed upper limbs in this study design is a limitation despite being a clinically focused point. Mentioning this in the title and the study objective will help your readership to focus on the target audience.

Response to comment 2

We appreciate your helpful suggestions. We have revised our title and research objectives.

“Feature analysis of joint motion in paralyzed and non-paralyzed upper limbs while reaching the occiput: a cross-sectional study in patients with mild hemiplegia” (Title, lines 4–5)

“This cross-sectional study aimed to clarify in patients with mild hemiplegia the kinematic characteristics of paralyzed and non-paralyzed upper limbs reaching the occiput.” (Abstract, lines 29–31)

“Based on the above discussion, this study aimed to clarify the kinematic characteristics of paralyzed and non-paralyzed upper limbs in patients with chronic phase mild hemiplegia while reaching the occipital region with the hand using 3DMA.” (Introduction, lines 119–121)

Comment 3

L158–161, P7: In G*power, I find this only in the exact test, and the effect size “g” is displayed there. What test family was used in your calculation? Also, what was your required sample size? 

Response to comment 3

We appreciate your remarks regarding sample size. We have corrected an error in our description of how we calculated the sample size using G*power. For the G*power calculation, the selected test family was “Exact”, and the sample size was calculated to be eight patients. The results of the sample size calculation are presented in a screenshot of the letter file. 

“The sample size was calculated by setting the difference from the constant (test family: exact test, binomial test, one sample case). For calculating the required sample size, the effect size (g) was 0.4, α was 0.05, power (1-β) was 0.8, and the constant proportion was 0.5.” (Materials and methods, Sample size, lines 161–164)

Comment 4

L185, P8: In this figure caption, please explain the landmark.

Response to comment 4

We appreciate your suggestion. We have added a description of the placed landmark to the caption of Fig 2.

“Fig 2. Measurement environment. (a) Side view and (b) top view. A landmark was placed 5 m away from the participants. By looking at it, neck and trunk movement was minimized in participants performing the reaching task.” (Materials and methods, Experimental procedure, lines 188–190)

Comment 5

L223, P9: For illustrations in each trial, left is outward, and right is return motions. This should be added to the figure caption.

Response to comment 5

We refer to your comments and have added an explanation of the illustration to the caption of Fig 4.

“Fig 4. Measurement sequence. The illustrations on the left of each trial show the outward motion from the starting position until the hand reaches the back of the head, and the illustrations on the right show the return motion from the point where the hand reaches the back of the head to the original position.” (Materials and methods, Reaching task, lines 229–232)

Comment 6

L246–251, P10–11: Did you input 50 data points into descriptive statistics and statistical tests? In general, each participant’s data is averaged first, then descriptive statistics are calculated, after which statistical tests are employed. In general, averaging trial data for each participant is performed to correct measurement errors. This is followed by statistical tests, which are performed to estimate the population.

Response to comment 6

Thank you for your valuable feedback on our statistical approach regarding the analysis of data points in our manuscript. In our analysis, descriptive statistics, multivariate analysis of covariance (MANCOVA), and random forest analyses were performed on 50 data (10 patients × 5 trials) for each paralyzed and non-paralyzed side. We did not analyze the data as 10 data averaged over 5 trials per person. We understand your concern about the typical process of averaging individual participant data before conducting further statistical analyses. This method is indeed a common approach to minimize variability due to measurement errors and to ensure that the analysis reflects the central tendency of each participant’s performance.

In our study, the approach to analyzing individual data points without first averaging them was intentional and is justified by the following considerations: 

1. Individual Variation Exploration: Our study aimed not only to explore general trends but also to capture individual variations in motor performance across trials. This was particularly important as we hypothesized that there might be significant trial-to-trial variability in motor task performance among stroke survivors, which could be clinically relevant.

2. Data Structure and Analysis Technique: We input 50 data points directly into the descriptive and inferential statistical analyses to maintain the richness of the data, which includes within-subject variability. This approach allowed us to apply more complex models that consider both within-subject (repeated measures) and between-subject variability simultaneously, enhancing the robustness of our findings.

3. Statistical Rigor: We employed statistical techniques that are robust to the inclusion of multiple measurements from the same participants, such as MANCOVA and mixed-effects models, which inherently adjust for the non-independence of repeated measurements within subjects.

However, we acknowledge that averaging data prior to analysis is a common practice and can be beneficial in simplifying the data presentation and interpretation. We have added the following explanation of data averaging to the manuscript.

“Data from the same participant were not averaged to allow subsequent analyses using random forest clustering.” (Materials and methods, Data acquisition and analysis, lines 260–261) 

Comment 7

L301, P13: Provide the effect size calculation and eligibility criteria.

Response to comment 7

Our description of the effect size calculation was indeed missing. Based on your comments, we have added the following statement to the manuscript.

“The η2 was used for the effect size in the statistical analysis, and the effect size indices were set as 0.01 for small, 0.06 for medium, and 0.14 for large [46].” (Materials and methods, Statistical analysis, lines 318–320)

“Reference #46 Cohen J. Statistical power analysis for the behavioral sciences. 2nd ed. New Jersey: L. Erlbaum Associates; 1988.”

Comment 8

L307, P13: To my knowledge, MANCOVA is used to compare independent samples, which means it is not applicable here.

Response to comment 8

Thank you for your constructive comment regarding the MANCOVA use in our study. We acknowledge the concern that MANCOVA is typically used for comparing independent samples. In our study, we employed MANCOVA to assess differences in joint motion characteristics between paralyzed and non-paralyzed limbs within subjects who have undergone stroke rehabilitation. This method was chosen to control for potential confounding variables such as age, sex, and body mass index, which might influence the motor outcomes. The following recently published papers use the MANCOVA (within-subjects design) analysis with repeated measures as in the present study.

Thompson CM, Voorhees HL, Taniguchi-Dorios E, Makos S, Pool K, Babu S. Development and initial assessment of an emotional support provision training intervention for interpersonal support providers in the context of chronic illness. Health Commun. 2024 Mar 11:1-14. doi: 10.1080/10410236.2024.2325183.

Figueroa-Padilla I, Rivera Fernández DE, Cházaro Rocha EF, Eugenio Gutiérrez AL, Jáuregui-Renaud K. Body weight may have a role on neuropathy and mobility after moderate to severe COVID-19: an exploratory study. Medicina (Kaunas). 2022 Oct 6;58(10):1401. doi: 10.3390/medicina58101401.

We recognize that the use of MANCOVA in a repeated measures context (within-subject design) is less typical and might have led to confusion. Our intention was to exploit the multivariate capability of MANCOVA to handle multiple dependent variables—motor time, joint angles, and angular velocities—which are not independent of each other.

We appreciate the opportunity to improve the clarity and accuracy of our analysis, and we have revised our manuscript to better justify our choice of statistical methods or adapt our approach in line with best statistical practices.

“Repeated measures multivariate analysis of covariance was performed on these data.” (Abstract, line 36)

“A repeated measures multivariate analysis of covariance (within-subject design) was performed to test this hypothesis.” (Materials and methods, Statistical analysis, lines 313–314)

“Binomial logistic regression analysis was conducted on the features fitted to the model for the paralyzed and non-paralyzed sides using repeated measures multivariate analysis of covariance.” (Materials and methods, Statistical analysis, lines 322–324)

“Repeated measures multivariate analysis of covariance was performed.” (Results, Detection of paralyzed upper limb with motion data, line 412)

“Binomial logistic regression analysis of the model-fitted features on the paralyzed and non-paralyzed sides was performed using repeated measures multivariate analysis of covariance.” (Results, Detection of paralyzed upper limb with motion data, lines 429–431)

“A repeated measures multivariate analysis of covariance was performed to test the hypothesis that motor time, joint angles, and angular velocity while doing movements to reach the occiput differ between paralyzed and non-paralyzed upper limbs.” (Discussion, lines 517–519)

“Next, to detect the cutoff values of the motor features that discriminate between the paralyzed and non-paralyzed sides, binomial logistic regression analysis was conducted on the features fitted to the models for the paralyzed and non-paralyzed sides using repeated measures multivariate analysis of covariance.” (Discussion, lines 525–528)

We appreciate all the comments and suggestions provided in order to improve our manuscript, and we sincerely thank all the reviewers.

---

## [Editor Report · Decision Letter 2]

6 May 2024

Feature analysis of joint motion in paralyzed and non-paralyzed upper limbs while reaching the occiput: a cross-sectional study in patients with mild hemiplegia

PONE-D-23-37600R2

Dear Dr. Hamaguchi,

We’re pleased to inform you that your manuscript has been judged scientifically suitable for publication and will be formally accepted for publication once it meets all outstanding technical requirements.

Kind regards,

Andrea Tigrini, Ph.D.

Academic Editor

PLOS ONE

Additional Editor Comments (optional):

Authors provider a very detailed study and all the concerns were addressed.
---

## [Editor Report · Acceptance letter]

13 May 2024

PONE-D-23-37600R2 

PLOS ONE

Dear Dr. Hamaguchi, 

I'm pleased to inform you that your manuscript has been deemed suitable for publication in PLOS ONE. Congratulations! Your manuscript is now being handed over to our production team.

Kind regards, 

on behalf of

Dr. Andrea Tigrini 

Academic Editor

PLOS ONE